# Allosteric modulation of GPCR-induced β-arrestin trafficking and signaling by a synthetic intrabody

Mithu Baidya[1,7], Madhu Chaturvedi[1,7], Hemlata Dwivedi-Agnihotri[1,7], Ashutosh Ranjan[1], Dominic Devost [2], Yoon Namkung [3], Tomasz Maciej Stepniewski[4,5], Shubhi Pandey[1], Minakshi Baruah[1], Bhanupriya Panigrahi[1], Parishmita Sarma[1], Manish K. Yadav[1], Jagannath Maharana[1], Ramanuj Banerjee[1], Kouki Kawakami[6], Asuka Inoue[6], Jana Selent[4], Stéphane A. Laporte [2,3], Terence E. Hébert[2] & Arun K. Shukla [1] ✉

Agonist-induced phosphorylation of G protein-coupled receptors (GPCRs) is a primary determinant of β-arrestin (βarr) recruitment and trafficking. For several GPCRs such as the vasopressin receptor subtype 2 (V₂R), agonist-stimulation first drives the translocation of βarrs to the plasma membrane, followed by endosomal trafficking, which is generally considered to be orchestrated by multiple phosphorylation sites. We have previously shown that mutation of a single phosphorylation site in the V₂R (i.e., V₂R^T360A) results in near-complete loss of βarr translocation to endosomes despite robust recruitment to the plasma membrane, and compromised ERK1/2 activation. Here, we discover that a synthetic intrabody (Ib30), which selectively recognizes activated βarr1, efficiently rescues the endosomal trafficking of βarr1 and ERK1/2 activation for V₂R^T360A. Molecular dynamics simulations reveal that Ib30 enriches active-like βarr1 conformation with respect to the inter-domain rotation, and cellular assays demonstrate that it also enhances βarr1-β₂-adaptin interaction. Our data provide an experimental framework to positively modulate the receptor-transducer-effector axis for GPCRs using intrabodies, which can be potentially integrated in the paradigm of GPCR-targeted drug discovery.

G protein-coupled receptors (GPCRs) recognize a broad spectrum of ligands and play critical roles in nearly every aspect of human physiology[1,2], and these receptors continue to be a major class of targets for novel drug discovery[3]. The spatio-temporal aspects of GPCR signaling are tightly regulated by multifunctional proteins, β-arrestins (βarrs)[4,5], and agonist-induced phosphorylation of GPCRs is a key determinant of βarr interaction and their ensuing functional outcomes[6,7]. While some GPCRs interact transiently with βarrs at the plasma membrane followed by rapid dissociation, others display a prolonged interaction resulting in endosomal trafficking of receptor-

[1]Department of Biological Sciences and Bioengineering, Indian Institute of Technology, Kanpur 208016, India. [2]Department of Pharmacology and Therapeutics, McGill University, Montréal, QC H3G 1Y6, Canada. [3]Department of Medicine, McGill University Health Center, McGill University, Montréal, QC H4A 3J1, Canada. [4]Research Program on Biomedical Informatics (GRIB), Department of Experimental and Health Sciences of Pompeu Fabra University (UPF)-Hospital del Mar Medical Research Institute (IMIM), 08003 Barcelona, Spain. [5]Faculty of Chemistry, Biological and Chemical Research Centre, University of Warsaw, Warsaw, Poland. [6]Graduate School of Pharmaceutical Sciences, Tohoku University, Sendai, Miyagi 980-8578, Japan. [7]These authors contributed equally: Mithu Baidya, Madhu Chaturvedi, Hemlata Dwivedi-Agnihotri. ✉e-mail: arshukla@iitk.ac.in

βarr complexes[8]. These two patterns of βarr interaction and trafficking have been used to categorize corresponding receptors as class A or class B GPCRs, respectively[8]. Interestingly, distinct phosphorylation patterns on GPCRs have been linked to different βarr conformations, which in turn determine the resulting functional responses[9–11]. While cumulative phosphorylation on GPCRs is typically believed to determine the affinity of βarr interaction, emerging evidence now suggests that spatial positioning of even single phosphorylation sites may provide a decisive contribution to βarr recruitment and subsequent functional outcomes[10–12].

We previously reported that mutation of a single phosphorylation site in the vasopressin receptor subtype 2 ($V_2R$) at $Thr^{360}$ in the carboxyl-terminus (i.e., $V_2R^{T360A}$) dramatically altered βarr trafficking patterns[10]. $V_2R$ is a class B receptor in terms of βarr interaction and trafficking where agonist stimulation first results in membrane recruitment of βarrs, followed by endosomal co-localization[8]. Interestingly, upon agonist-stimulation of $V_2R^{T360A}$, βarrs efficiently translocate to the plasma membrane, but do not traffic to endosomal compartments, unlike the wild-type receptor even after prolonged agonist-exposure (Fig. 1a)[10]. $V_2R^{T360A}$ also exhibits reduced levels of ERK1/2 activation compared to the wild-type receptor without any measurable effect on G protein-coupling as assessed by measuring cAMP production[10]. This mutation leads to the disruption of a salt-bridge with $Lys^{294}$ in βarr1 and consequently reduces the fraction of active βarr1 conformation as assessed using molecular dynamics simulation[10]. While $V_2R^{T360A}$ exhibits a dramatic alteration in βarr1 trafficking pattern, the other phospho-site mutants behave either like wild-type (e.g. $V_2R^{S357A}$ and $V_2R^{T359A}$) or exhibit a near-complete loss of βarr1 translocation to the membrane (e.g. $V_2R^{S362A/S363A/S364A}$)[10]. This prompted us to probe the conformation of βarr1 in the context of this receptor mutant ($V_2R^{T360A}$) and compare it with the wild-type $V_2R$, using a previously described intrabody30 (Ib30) based sensor[13].

Here, we show that Ib30 robustly recognizes βarr1 recruited to the plasma membrane upon agonist-stimulation of $V_2R^{T360A}$, and it rescues endosomal localization of βarr1 and ERK1/2 activation for $V_2R^{T360A}$ to the levels of the wild-type receptor. We also discover that Ib30 enriches an active-like conformational population of βarr1 and also enhances the interaction of βarr1 with $β_2$-adaptin. These findings establish the capability of Ib30 to allosterically modulate βarr1 trafficking and activation for $V_2R^{T360A}$, and potentially open a paradigm to modulate GPCR signaling using designer allosteric modulators.

## Results

### βarr1 conformation induced by $V_2Rpp^{T360}$ phospho-peptides

Synthetic phospho-peptides corresponding to the carboxyl-terminus of $V_2R$ have previously been used to activate βarrs in-vitro and probe the activation-induced conformational changes[14,15]. Therefore, we first synthesized two phospho-peptides corresponding to $V_2R^{T360}$ mutation to probe whether the absence of $Thr^{360}$ phosphorylation influences βarr1 conformation. These two phospho-peptides, referred to as $V_2Rpp^{T360-1}$ and $V_2Rpp^{T360-2}$, contain a non-phosphorylated threonine (T) or alanine (A) at position 360, respectively, while the rest of the sequence and phosphorylation patterns are identical to $V_2Rpp$ (referred to as $V_2Rpp^{WT}$) (Fig. 1b). We used a previously described limited trypsin proteolysis assay[15] to compare βarr1 conformation induced by $V_2Rpp^{T360}$ phospho-peptides with that of $V_2Rpp^{WT}$. We observed that the activation of βarr1 by $V_2Rpp^{WT}$ resulted in an accelerated cleavage of the 48 kDa band ($Gly^{-8}-Arg^{418}$), protection of 47 kDa and 45 kDa bands ($Leu^1-Arg^{418}$ and $Leu^1-Arg^{393}$, respectively) and appearance of a 21 kDa band ($Leu^1-Arg^{188}$) (Fig. 1c, d, Supplementary Fig. 1a, b) as reported previously[15]. Interestingly, $V_2Rpp^{T360}$ phospho-peptides also induced a proteolysis pattern qualitatively similar to that observed for $V_2Rpp^{WT}$, although there were noticeable differences such as relatively slower proteolysis of the 48 kDa band and a weaker intensity of the 21 kDa band (Fig. 1c, d and Supplementary Fig. 1a, b). The difference in

the intensity of 48 kDa and 47 kDa bands are visualized better at 1:50 ratio of trypsin:βarr1 (Fig. 1c, right half), while the difference in the 32 kDa and 21kda bands are visualized better at 1:25 ratio (Fig. 1c, left half). This observation indicates that $V_2R^{T360}$ phospho-peptides are capable of binding βarr1; however, they do not induce active βarr1 conformation as stabilized by $V_2R^{WT}$ but instead appear to promote an intermediate state between the basal and active-conformations.

### Fab30/ScFv30 sensors recognize $V_2Rpp^{T360}$-βarr1 complexes

As an additional readout of βarr1 conformation induced by $V_2Rpp^{WT}$ vs. $V_2Rpp^{T360}$ phospho-peptides, we measured the ability of antibody fragments referred to as Fab30/ScFv30 to recognize $V_2Rpp$-βarr1 complexes using co-immunoprecipitation (co-IP). Fab30 and ScFv30 are known to selectively recognize βarr1 conformation induced by $V_2Rpp^{WT}$, and thus, they have been used previously as conformational biosensors to monitor βarr activation in vitro[13,16]. We observed that Fab30/ScFv30 robustly interacted with the $V_2Rpp^{T360-1/2}$-βarr1 complexes, albeit at lower levels than $V_2Rpp^{WT}$ (Fig. 2a–d). We carried out co-IP in the presence of either 10-fold or 50-fold molar excess of the phospho-peptides compared to βarr1, but the reactivity patterns of Fab30/ScFv30 did not change significantly (Fig. 2a–d). Similar to the limited proteolysis data presented in Fig. 1, these data also suggest that $V_2Rpp^{T360}$ phospho-peptides induce a conformation in βarr1, which is qualitatively similar to that of $V_2Rpp^{WT}$, but not identical. However, we cannot rule out the possibility that the binding affinities of Fab30 and ScFv30 for βarr1-$V_2Rpp^{T360-1/2}$ complexes are relatively lower compared to βarr1-$V_2Rpp^{WT}$ complex, which requires additional experimentation.

Considering the patterns of limited proteolysis and Fab30/ScFv30 reactivity induced by these phospho-peptides, we next carried out limited proteolysis assays in the presence of ScFv30 (Fig. 2e, f, Supplementary Fig. 2). We observed that the 47 kDa band ($Leu^1-Arg^{418}$) was significantly protected in presence of ScFv30, and the bands at 32 kDa and 21 kDa ($Leu^1-Arg^{285}$ and $Leu^1-Arg^{188}$, respectively) did not appear (Fig. 2e, f, Supplementary Fig. 2). Interestingly, the proteolysis patterns observed in presence of ScFv30 were nearly-identical for $V_2Rpp^{WT}$ and $V_2Rpp^{T360-1/2}$ phospho-peptides, although an additional band at ~30 kDa was observed only with $V_2Rpp^{WT}$ (Fig. 2e, f). The converging proteolysis patterns of βarr1 observed in the presence of ScFv30 for the wild-type and mutant peptides suggest that ScFv30 might be promoting the transition of $V_2Rpp^{T360}$-bound βarr1 conformation towards the active-like state.

### Structural insights into βarr1 conformation induced by $V_2Rpp^{WT}$ and $V_2Rpp^{T360}$

Taking a lead from the limited proteolysis assays, we next analyzed the crystal structures of βarr1 in basal, $V_2Rpp^{WT}$-, and $V_2Rpp^{T360}$-bound states to gain further insights into βarr1 conformation. As the distal carboxyl-terminus of βarr1 is not resolved in these structures, we focused primarily on $Arg^{285}$ and $Arg^{188}$, which are the trypsin cleavage sites yielding the 32 kDa ($Leu^1-Arg^{285}$) and 21 kDa ($Leu^1-Arg^{188}$) bands, respectively. Both of these residues exhibit a reorientation of their side chains between basal and phospho-peptide-bound conformations (Fig. 3a). The network of interactions involving $Arg^{285}$ and $Arg^{188}$ are also mostly maintained between basal and peptide-bound conformations although there are some differences as well (Fig. 3a). Further analysis of the local interaction networks of $Arg^{188}$ and $Arg^{285}$ using CONTACT/ACT program within the CCP4 suite[17], which analyzes all possible contacts/interactions and distances between residues in protein structures, including water molecules within a specified distance, also converges to the same observation as evident from the crystal structures (Supplementary Fig. 3). We also carried out molecular dynamics (MD) simulation studies using the crystal structures of βarr1 as templates to probe the conformational ensemble sampled by $Arg^{188}$ and $Arg^{285}$, and observed that they explore similar conformational space in the wild-type and mutant phospho-peptide-bound

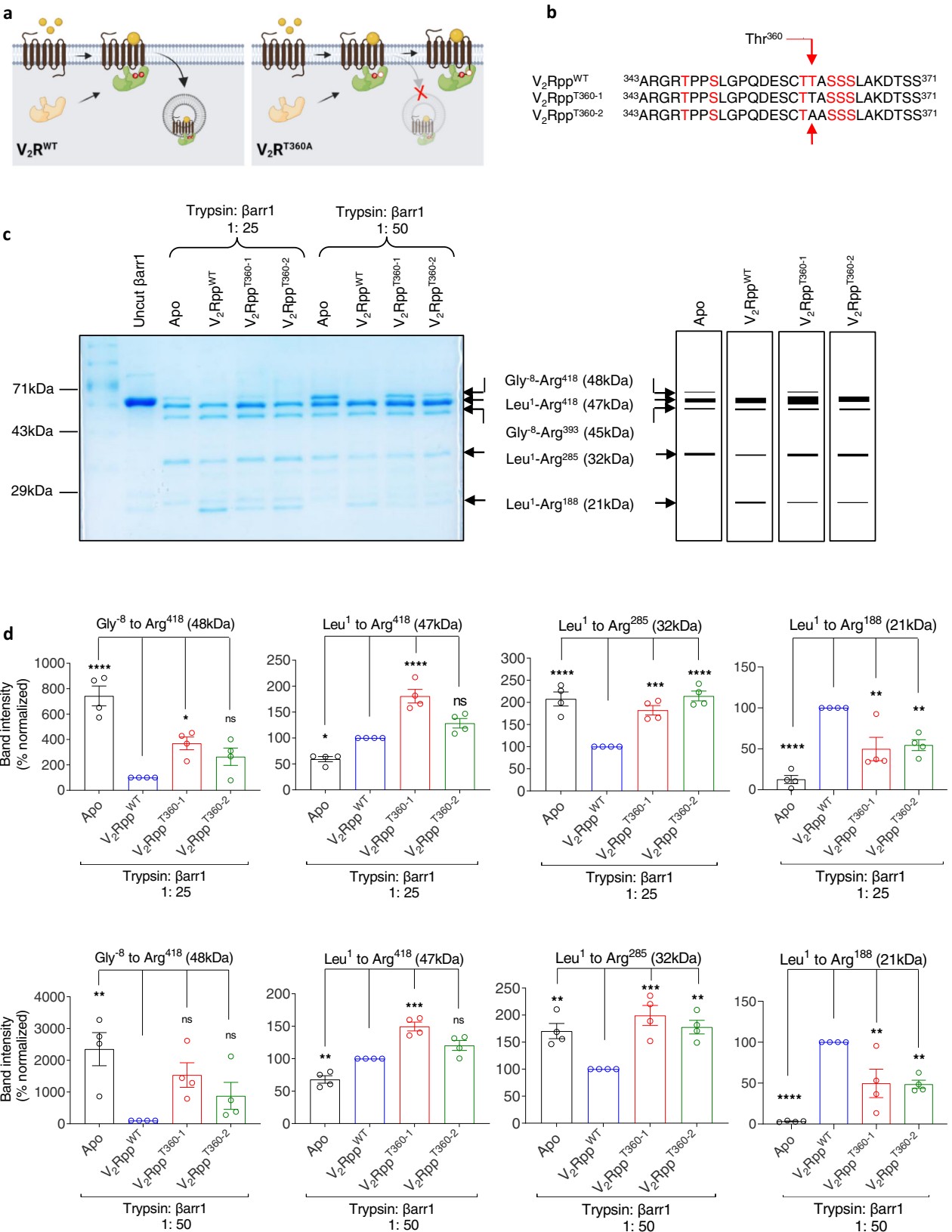

states (Fig. 3b). Taken together, these structural insights provide a plausible explanation for the proteolysis patterns obtained for $V_2Rpp^{WT}$ vs. $V_2Rpp^{T360}$, and support the hypothesis that $V_2Rpp^{T360}$ induces an intermediate conformation in βarr1 compared to apo- and $V_2R^{WT}$-bound states that may be further influenced by the binding of Fab30.

## Agonist-induced βarr1 recruitment to $V_2R^{WT}$ and $V_2R^{T360A}$

The experiments presented so far were carried out in-vitro using isolated phospho-peptides, and therefore, we next set out to measure the relative recruitment of βarr1 and the reactivity of Ib30, an intrabody derived from Fab30 that reports βarr1 activation and trafficking[11,13], for the wild-type $V_2R$ ($V_2R^{WT}$) and the mutant receptor ($V_2R^{T360A}$). We first

**Fig. 1 | V2Rpp$^{WT}$ and V2Rpp$^{T360}$ impart different conformations on βarr1.**
**a** Schematic representation of the inability of V$_2$R$^{T360A}$ mutant to promote endosomal trafficking of βarr1 as published previously. **b** Amino acid sequences of the V$_2$R$^{WT}$ and Thr$^{360}$ mutant phospho-peptides (V$_2$Rpp$^{T360-1}$ and V$_2$Rpp$^{T360-2}$) were used in this study. Phosphorylated residues (serine, S; threonine, T) are highlighted in red, and the position 360 is indicated by an arrow. V$_2$Rpp$^{T360-1}$ and V$_2$Rpp$^{T360-2}$ contain a non-phosphorylated threonine and an alanine at position 360, respectively. **c** Limited trypsin proteolysis of βarr1 (5 min) in the absence or presence of indicated phospho-peptides at two different trypsin: βarr1 ratio followed by visualization of the bands on SDS-PAGE. A representative gel from four independent experiments (left panel) and a schematic of the proteolysis pattern corresponding to 1: 25 ratio of trypsin:βarr1 (right panel) is shown here. **d** Densitometry-based quantification (mean ± SEM) of indicated bands from four independent experiments, normalized

with respect to V$_2$Rpp$^{WT}$ condition (treated as 100%) (One-way ANOVA, Dunnett's multiple comparisons test). The exact $p$ values are as follows: Gly$^{-8}$ to Arg$^{418}$ (48 kDa) band (1: 25)- Apo ($p < 0.0001$), V$_2$R$^{T360-1}$ ($p = 0.0159$), V$_2$R$^{T360-2}$ ($p = 0.1539$); Gly$^{-8}$ to Arg$^{418}$ (48 kDa) band (1: 50)- Apo ($p = 0.0039$), V$_2$R$^{T360-1}$ ($p = 0.0566$), V$_2$R$^{T360-2}$ ($p = 0.385$); Leu$^1$ to Arg$^{418}$ (47 kDa) band (1: 25)-Apo ($p = 0.0132$), V$_2$R$^{T360-1}$ ($p < 0.0001$), V$_2$R$^{T360-2}$ ($p = 0.0844$); Leu$^1$ to Arg$^{418}$ (47 kDa) band (1: 50)-Apo ($p < 0.0024$), V$_2$R$^{T360-1}$ ($p = 0.0001$), V$_2$R$^{T360-2}$ ($p = 0.059$); Leul$^1$ to Arg$_2^{85}$ (32 kDa) band (1: 25)-Apo ($p < 0.0001$), V$_2$R$^{T360-1}$ ($p = 0.0006$), V$_2$R$^{T360-2}$ ($p < 0.0001$); Leul$^1$ to Arg$^{285}$ (32 kDa) band (1: 50)-Apo ($p = 0.0072$), V$_2$R$^{T360-1}$ ($p = 0.0005$), V$_2$R$^{T360-2}$ ($p = 0.0036$); Leul$^1$ to Arg$^{188}$ (21 kDa) band (1: 25)-Apo ($p < 0.0001$), V$_2$R$^{T360-1}$ ($p = 0.0028$), V$_2$R$^{T360-2}$ ($p = 0.0057$); Leul$^1$ to Arg$^{188}$ (21 kDa) band (1: 50)-Apo ($p < 0.0001$), V$_2$R$^{T360-1}$ ($p = 0.005^1$), V$_2$R$^{T360-2}$ ($p = 0.0044$). Source data are provided as a Source Data file (*$p < 0.05$, **$p < 0.01$, ***$p < 0.001$, ****$p < 0.0001$, ns = non-significant).

used a NanoBiT assay that measures the direct binding of the receptor and βarr1 and, therefore, reports cumulative interaction resulting from both the cell surface and internalized pools (Fig. 4a). We observed that the total βarr1 recruitment to V$_2$R$^{T360A}$ mutant was significantly attenuated compared to the V$_2$R$^{WT}$ (Fig. 4b), and this is in excellent agreement with our previous study using the Tango assay[10]. Next, we compared the surface recruitment of βarr1 to the wild-type and mutant receptor using a NanoBiT assay where the LgBiT component is tethered to the plasma membrane through CAAX sequence while βarr1 is tagged with SmBiT (Fig. 4c). Here, we observed a near-identical pattern of βarr1 recruitment for the V$_2$R$^{WT}$ and V$_2$R$^{T360A}$ suggesting that endosomal trafficking but not surface translocation is compromised by the Thr$^{360}$Ala mutation in the V$_2$R (Fig. 4d). These findings, therefore, set the stage for testing whether Ib30 can recognize the βarr1 conformation induced by V$_2$R$^{T360A}$ and influence its endosomal trafficking. In these experiments, surface expression of the V$_2$R$^{WT}$ and V$_2$R$^{T360A}$ were comparable as measured using whole-cell ELISA assay (Supplementary Fig. 4a, b).

**Intrabody30 rescues endosomal trafficking of βarr1 for V$_2$R$^{T360A}$**
We first co-expressed SmBiT-βarr1 and LgBiT-Ib30 constructs with V$_2$R$^{WT}$ and V$_2$R$^{T360A}$ and measured agonist-induced changes in luminescence signal as a readout of βarr1-Ib30 interaction and conformational recognition of βarr1 by Ib30 (Fig. 5a). As Ib30 reactivity is expected to follow cumulative βarr1 recruitment upon receptor activation, we anticipated a relatively lower response for V$_2$R$^{T360A}$ compared to V$_2$R$^{WT}$ considering their total βarr1 recruitment patterns as presented in Fig. 4b. Surprisingly, however, we observed nearly-identical response for Ib30 reactivity upon agonist-stimulation for both, V$_2$R$^{WT}$ and V$_2$R$^{T360A}$ (Fig. 5b). This finding not only suggests that the conformation of βarr1 induced by V$_2$R$^{T360A}$ in the cellular context is recognizable by the Ib30 sensor, but also that Ib30 might be rescuing endosomal trafficking of βarr1 and thereby, bringing the overall recruitment to the wild-type level. We tested this hypothesis by measuring the overall βarr1 recruitment for V$_2$R$^{WT}$ and V$_2$R$^{T360A}$ in a NanoBiT assay in presence of either a control intrabody (Ib-CTL) or Ib30. In fact, we observed that overall βarr1 recruitment for V$_2$R$^{T360A}$ becomes nearly-identical to that of V$_2$R$^{WT}$ upon co-expression of Ib30 (Supplementary Fig. 5). In these NanoBiT experiments, the V$_2$R$^{WT}$ and V$_2$R$^{T360A}$ were expressed at comparable levels as measured in terms of their surface expression (Supplementary Fig. 4c, d). We also measured agonist-induced G protein-coupling for the V$_2$R$^{T360A}$ in the presence of Ib-CTL and Ib-30 but did not observe any significant difference, similar to the V$_2$R$^{WT}$ (Supplementary Fig. 6a, b), suggesting the specificity of Ib30 for receptor-βarr interaction without a measurable effect on G proteincoupling.

In order to directly visualize the ability of Ib30 to recognize βarr1 upon recruitment to V$_2$R$^{T360A}$, we co-expressed Ib30-mYFP construct together with βarr1-mCherry in HEK-293 cells expressing V$_2$R$^{T360A}$, and monitored localization of βarr1 and Ib30 by confocal microscopy. Ib30

translocated to the plasma membrane upon agonist-stimulation, similar to βarr1, and exhibited robust colocalization with βarr1 (Fig. 5c, d). Interestingly, we also observed that upon prolonged agonist exposure (>15 min), both βarr1 and Ib30 were translocated to endosomal vesicles and robustly co-localized (Fig. 5c, d). This observation further strengthens the hypothesis that Ib30 may potentially be rescuing endosomal trafficking of βarr1 for V$_2$R$^{T360A}$ as hinted in the NanoBiT-based Ib30 recognition assay (Fig. 5a, b) and overall βarr1 recruitment assay for V$_2$R$^{T360A}$ (Supplementary Fig. 5). To further corroborate these findings and directly establish the allosteric potentiation of endosomal trafficking of βarr1 by Ib30, we used three different approaches. First, we co-expressed a βarr1-mYFP construct in HEK-293 cells together with either V$_2$R$^{WT}$ or V$_2$R$^{T360A}$ in the presence or absence of HA-tagged Ib30. We monitored the localization of βarr1 in these cells upon agonist-simulation using confocal microscopy and scored the localization pattern of βarr1 in terms of plasma membrane vs. internalized vesicles. We manually scored more than 500 cells for each condition and plotted the data as % normalized (i.e. % of total cells displaying membrane vs. punctate localization of βarr1). In line with data presented in Fig. 5c, d, we observed that the presence of Ib30 indeed promoted endosomal trafficking of βarr1 for V$_2$R$^{T360A}$ (Fig. 5e, f) while βarr1 remained localized primarily at the plasma membrane even after prolonged agonist-exposure in the absence of Ib30, as reported previously[10].

Next, we used an intermolecular bystander BRET assay to monitor the endosomal localization of βarr1 quantitatively by using βarr1-R-Luc and GFP-FYVE constructs, described previously (Fig. 6a)[18]. As shown in Fig. 6b, we observed a very low level of agonist-induced BRET for V$_2$R$^{T360A}$ in the presence of control intrabody (Ib-CTL), while V$_2$R$^{WT}$ exhibited a robust response as expected. Interestingly, however, co-expression of Ib30 rescued the BRET signal (i.e., endosomal trafficking of βarr1) to almost the same level as V$_2$R$^{WT}$ (Fig. 6b, c). We also observed an enhanced E$_{max}$ in BRET assay for V$_2$R$^{WT}$ in the presence of Ib30, compared to Ib-CTL, although basal BRET was also higher and therefore, the change in BRET signal is significantly more pronounced for the V$_2$R$^{T360A}$ (Fig. 6c). Finally, we also carried out a similar experiment using the NanoBiT assay that measures endosomal trafficking of βarr1 based on similar principles as in BRET assay described above (Fig. 6d). We observed analogous potentiation of endosomal localization of βarr1 by Ib30 for V$_2$R$^{T360A}$ as in BRET assay while there was no significant change for V$_2$R$^{WT}$ (Fig. 6e, f). In these experiments, the surface expression of the V$_2$R$^{WT}$ and V$_2$R$^{T360A}$ were maintained at comparable levels to ensure that the observed differences did not arise from a difference in the expression level of the wild-type and mutant receptors (Supplementary Fig. 4e). Taken together with the confocal microscopy observations, these data establish that Ib30 recognizes V$_2$R$^{T360A}$-bound βarr1 conformation and allosterically potentiates its trafficking to endosomal vesicles and thereby, rescues the trafficking pattern of βarr1 making it similar to the wild-type receptor.

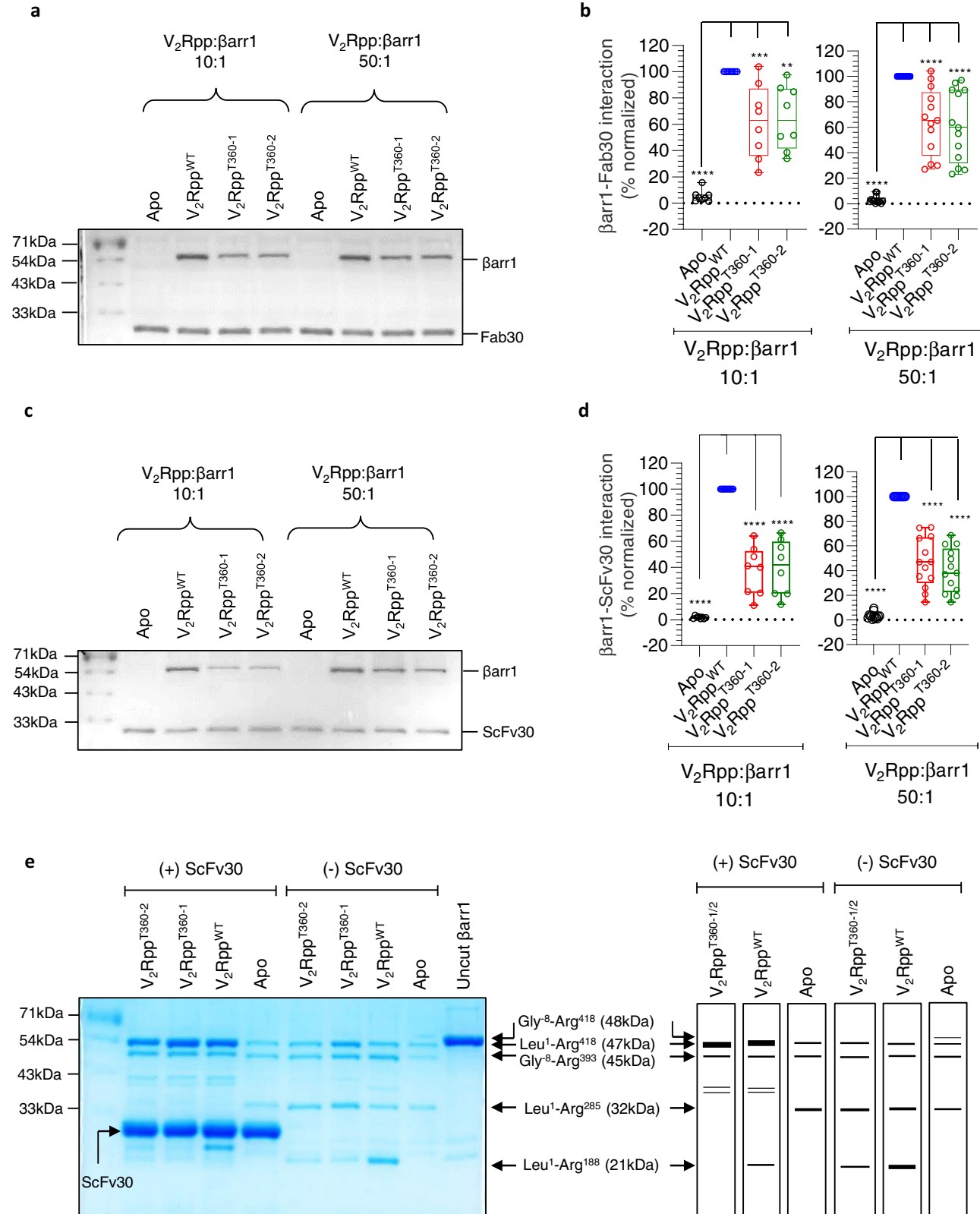

## Intrabody30 rescues agonist-induced ERK1/2 activation for V₂R^T360A

We have previously reported that agonist-induced ERK1/2 MAP kinase activation is significantly attenuated for $V_2R^{T360A}$ compared to the wild-type receptor[10]. Considering the potentiating effect of Ib30 on βarr1 trafficking, and previous studies linking endosomal pool of βarrs with ERK1/2 MAP kinase activation for GPCRs[19], we next measured the effect of Ib30 on agonist-induced ERK1/2 phosphorylation for the $V_2R^{WT}$ and $V_2R^{T360A}$. Expectedly, Ib30 did not have a significant effect on ERK1/2 activation for the $V_2R^{WT}$, however, it robustly enhanced the level of phosphorylated ERK1/2 upon agonist-stimulation for $V_2R^{T360A}$, to the levels of the $V_2R^{WT}$ (Fig. 7a, b). The surface expression of the wild-type

**Fig. 2 | Recognition of $V_2Rpp^{T360}$-induced βarr1 conformation by Fab30/ScFv30.**
**a**–**d**. Fab30 and ScFv30 recognize βarr1 conformation induced by $V_2Rpp^{T360}$ albeit less efficiently than that induced by $V_2Rpp^{WT}$. Purified βarr1 was incubated with the indicated phospho-peptides and Fab30/ScFv30 followed by co-immunoprecipitation (co-IP) using Protein L agarose and visualization using Coomassie-stained SDS-PAGE. A representative gel from eight to thirteen independent experiments is shown here. Panels **b** and **d** show densitometry-based quantification of βarr1-Fab30/ScFv30 interaction normalized with $V_2Rpp^{WT}$-βarr1 control (taken as 100%). The data is represented as box plots showing median, IQR with whiskers of 1.5× IQR, and circles representing values from independent experimental replicates. (One-way ANOVA, Dunnett's multiple comparisons test). The exact p values in **2b** are as follows: for ($V_2Rpp$:βarr1 10:1) Apo ($p < 0.0001$),

$V_2R^{T360-1}$ ($p = 0.001$), $V_2R^{T360-2}$ ($p = 0.0021$); for ($V_2Rpp$:βarr1 50:1) Apo, $V_2R^{T360-1}$, $V_2R^{T360-2}$ ($p < 0.0001$); The exact p values in **2d** are as follows: for ($V_2Rpp$:βarr1 10:1); Apo, $V_2R^{T360-1}$, $V_2R^{T360-2}$ ($p < 0.0001$); for ($V_2Rpp$:βarr1 50:1) Apo, $V_2R^{T360-1}$, $V_2R^{T360-2}$ ($p < 0.0001$); Source data are provided as a Source Data file (**$p < 0.01$ ***$p < 0.001$, ****$p < 0.0001$). **e** Binding of ScFv30 influences limited proteolysis pattern of βarr1 for the wild-type and mutant phospho-peptides similarly (30 min). βarr1 activated with 50-fold molar excess of indicated phospho-peptides was subjected to limited trypsin proteolysis at a trypsin:βarr1 ratio of 1: 50 in the presence or absence of ScFv30 followed by visualization of the bands on SDS-PAGE. A representative gel from four independent experiments (left panel) and a schematic of proteolysis patterns (right panel) are shown here. Source data are provided as a Source Data file.

and mutant receptors were at comparable levels in these experiments (Supplementary Fig. 4f). Taken together with the endocytosis data, these findings demonstrate an allosteric effect of Ib30 to positively modulate βarr-mediated functional responses for the $V_2R^{T360A}$ mutant in the cellular context.

## Structural insights into the allosteric effect of Fab30 on βarr1 conformation

The positive allosteric effect of Ib30 on endosomal trafficking of βarr1 and ERK1/2 activation for $V_2R^{T360A}$ prompted us to probe the potential structural mechanism for this phenomenon at the level of phospho-peptide binding and βarr1 conformation. Therefore, we first analyzed the crystal structures of $V_2Rpp^{WT}$-βarr1 (PDB: 4JQI) and $V_2Rpp^{T360-1}$-βarr1 (PDB: 7DFA), determined previously. Interestingly, a segment of the $V_2Rpp^{T360-1}$ containing residues Pro[353] to Thr[360] showed a marked repositioning compared to the $V_2Rpp^{WT}$ binding pose (Fig. 8a). In the $V_2Rpp^{WT}$-βarr1 crystal structure, pThr[360] engages Lys[294], Lys[11] and Arg[25] in βarr1 through ionic interactions, which is expectedly absent in case of $V_2Rpp^{T360}$ mutation. Of these, Lys[294] in the lariat loop and Lys[11] in the β-strand I of βarr1 are particularly noteworthy as they constitute a key part of the polar core and phosphate sensor, respectively[7]. These interactions are critical in the process of βarr1 activation upon binding of phosphorylated carboxyl-terminus of GPCRs. Interestingly, pThr[359] in $V_2Rpp^{T360}$ phospho-peptide engages with Lys[11] but not with Lys[294] or Arg[25]. This interesting structural rearrangement may, in part explain an intermediate active-like conformation induced by $V_2Rpp^{T360}$ phospho-peptides as observed in limited proteolysis and ScFv30 co-IP assay.

Next, we used molecular dynamics (MD) simulation on $V_2Rpp^{WT}$- and $V_2Rpp^{T360A}$-bound βarr1. We have previously reported that Thr[360]Ala mutation resulted in a significant shift in the population of βarr1 towards inactive-like conformation compared to the $V_2Rpp^{WT}$ as measured in terms of the inter-domain rotation angle[10]. We also found that the Thr[360]Ala mutation leads to the disruption of a salt-bridge with Lys[294] in the lariat loop of βarr1, which links the N- to the C-domain via the phospho-peptide, and removing this inter-domain connector may reverse the inter-domain rotation leading to the transition of βarr1 towards inactive conformation[10]. Now, in this study, we first reproduced this behavior for the $V_2Rpp^{T360A}$ mutant, demonstrating that introducing the Thr[360]Ala mutation in the $V_2Rpp$-βarr1 complex leads to a dramatic shift towards inactive-like conformations with an inter-domain rotation angles <15° (i.e. $V_2Rpp^{WT}$: 24% vs. $V_2Rpp^{T360A}$: 63%) (Fig. 8b–d). Strikingly, simulations of the $V_2Rpp^{T360A}$-βarr1 complex in presence of Fab30 revealed that Fab30 binding significantly stabilizes the population of active-like βarr1 conformations ($V_2Rpp^{T360A}$ + Fab30: 70% vs. $V_2Rpp^{T360A}$: 37%) (Fig. 8b–d). This interesting observation can be rationalized by the fact that Fab30 simultaneously binds to the N- and the C-domain of βarr1, which blocks the reversal of the inter-domain rotation towards inactive-like βarr1 conformations. This stabilizing contribution of Fab30 towards active-like βarr1 conformations may offer a plausible mechanism for the positive allosteric effect of

Ib30 observed on βarr1 trafficking to endosomes and agonist-induced ERK1/2 activation for the $V_2R^{T360A}$.

It is worth noting here that despite an accumulated simulation time of 10μs per system (i.e. 5 runs of 2 μs), our simulation set up does not allow for a converged sampling of βarr1 ensemble. However, the running averages of the inter-domain rotation angle show that βarr1 can transition between active- and inactive-like states in all three conditions (i.e. $V_2Rpp^{WT}$, $V_2Rpp^{T360A}$, and $V_2Rpp^{T360A}$ + Fab30). This indicates that βarr1 explores a wide range of conformational landscape and that observed tendencies are not an artifact but rather describe an actual property of each system.

## Intrabody30 enhances βarr1-β₂-adaptin interaction

Next, we set out to identify a potential functional correlate of Ib30-induced enrichment of active-like βarr1 conformation and to reveal the mechanism of Ib30-mediated endosomal targeting of βarr1. As the interaction of βarrs with $β_2$-adaptin is a prominent mechanism that drives GPCR endocytosis[5,20], we measured the effect of Ib30 on βarr1-$β_2$-adaptin interaction. We used the ear-domain of $β_2$-adaptin (592–951) tagged with GST at the N-terminus and assessed its interaction with βarr1 in the presence of lysate prepared from *Sf*9 cells expressing $V_2R^{T360A}$. The ear domain represents the C-terminal appendage of the $β_2$-adaptin subunit of the clathrin adaptor AP2 complex, and this region has been shown previously to interact with βarr1[21]. As presented in Fig. 9a, b, we observed a low but statistically significant interaction between βarr1 and $β_2$-adaptin in the presence of Ib-CTL. More interestingly, the presence of Ib30 enhanced this interaction several fold suggesting the ability of Ib30 to promote βarr1-$β_2$-adaptin interaction (Fig. 9a, b).

To further corroborate this interesting finding in cellular context, we next used a previously described BRET-based assay[22] to monitor agonist-induced interaction of βarr1 with $β_2$-adaptin in the presence of either Ib-CTL or Ib30 for $V_2R^{WT}$ and $V_2R^{T360A}$ (Fig. 9c). There was a robust interaction between βarr1 and $β_2$-adaptin for the $V_2R^{WT}$ upon agonist-stimulation in the presence of Ib-CTL, while the response was significantly lower for the $V_2R^{T360A}$. This is in line with significantly attenuated endosomal trafficking of βarr1 for the $V_2R^{T360A}$ compared to $V_2R^{WT}$. Interestingly, co-expression of Ib30 significantly enhanced βarr1-$β_2$-adaptin interaction for $V_2R^{T360A}$, bringing it to almost the same level as $V_2R^{WT}$. However, the basal BRET signal was also higher under Ib30 expression conditions compared to Ib-CTL, and it may reflect the propensity of Ib30 to enhance βarr1-$β_2$-adaptin interaction, even under the basal condition i.e. without receptor activation (Fig. 9d). To test this hypothesis, we carried out a titration experiment, where we expressed Ib30 at increasing levels and assessed βarr1-$β_2$-adaptin interaction in the BRET assay. As presented in Fig. 9e, increasing expression of Ib30 indeed enhanced BRET in a saturable manner suggesting the ability of Ib30 to promote basal interaction between βarr1 and $β_2$-adaptin. Taken together, these observations provide a mechanistic basis of Ib30-induced allosteric modulation of βarr1 trafficking pattern observed for $V_2R^{T360A}$.

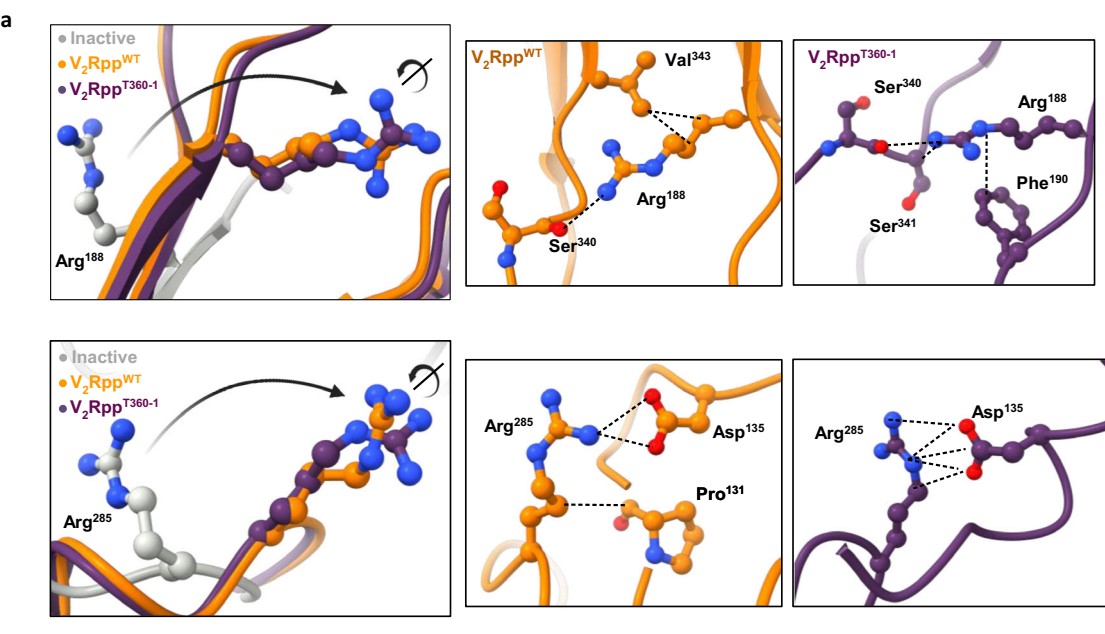

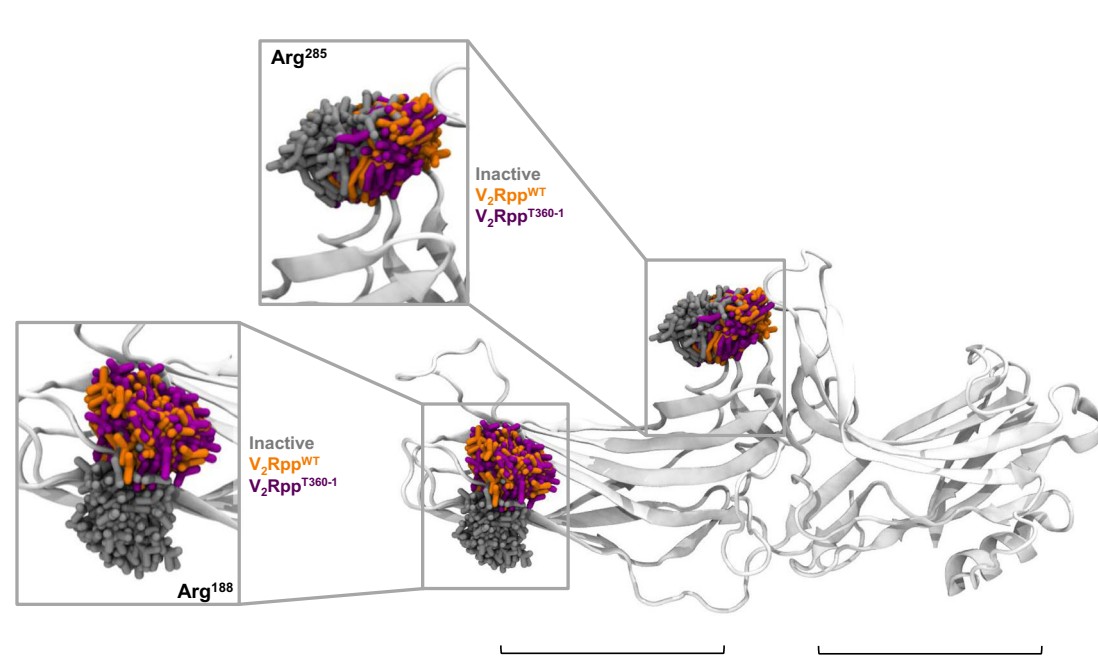

**Fig. 3 | Structural insights into binding of phospho-peptides to βarr1.**
**a** Structural snapshots comparing the relative orientation and local interaction networks of trypsin cleavage sites Arg[188] and Arg[285] in the crystal structures of βarr1 in basal (PDB: 1G4M, grey), V₂RppWT-bound (PDB: 4JQI, orange) and V₂RppT360-1-bound (PDB: 7DFA, violet) conformations. The dotted lines represent hydrogen bonds and polar interactions. **b** Molecular dynamics simulations based on the crystal structures confirm an overall similar conformational space sampled by Arg[188] and Arg[285], the two trypsin proteolysis sites which are protected by ScFv30.

## Discussion

In this study, we demonstrate that a synthetic intrabody (Ib30) can allosterically modulate agonist-induced trafficking patterns of βarr1 for a vasopressin receptor subtype 2 mutant (V₂RT360A) lacking a key phosphorylation site in its carboxyl-terminus. Ib30 induces the transition of βarr1 trafficking pattern for V₂RT360A from class A to class B by enriching the fraction of active-like conformational population of βarr1, and allosterically enhancing βarr1-β₂-adaptin interaction. Moreover, Ib30 also rescues the attenuated ERK1/2 activation for V₂RT360A to levels induced by the wild-type receptor. A

previous study has demonstrated a critical role of βarr-β₂-adaptin interaction in βarr-mediated ERK1/2 activation for V₂R using a small molecule inhibitor of this interaction[22]. Therefore, an increase in βarr1-β₂-adaptin interaction in presence of Ib30 may provide a plausible mechanism for its ability to rescue agonist-induced ERK1/2 activation for V₂RT360A. However, additional mechanisms may also contribute to this intriguing observation, and it would be interesting to probe this further in subsequent studies. Although Ib30 appears to slightly enhance the endosomal trafficking of βarr1 for V₂RWT (Fig. 6b, c), an observation that is consistent with our

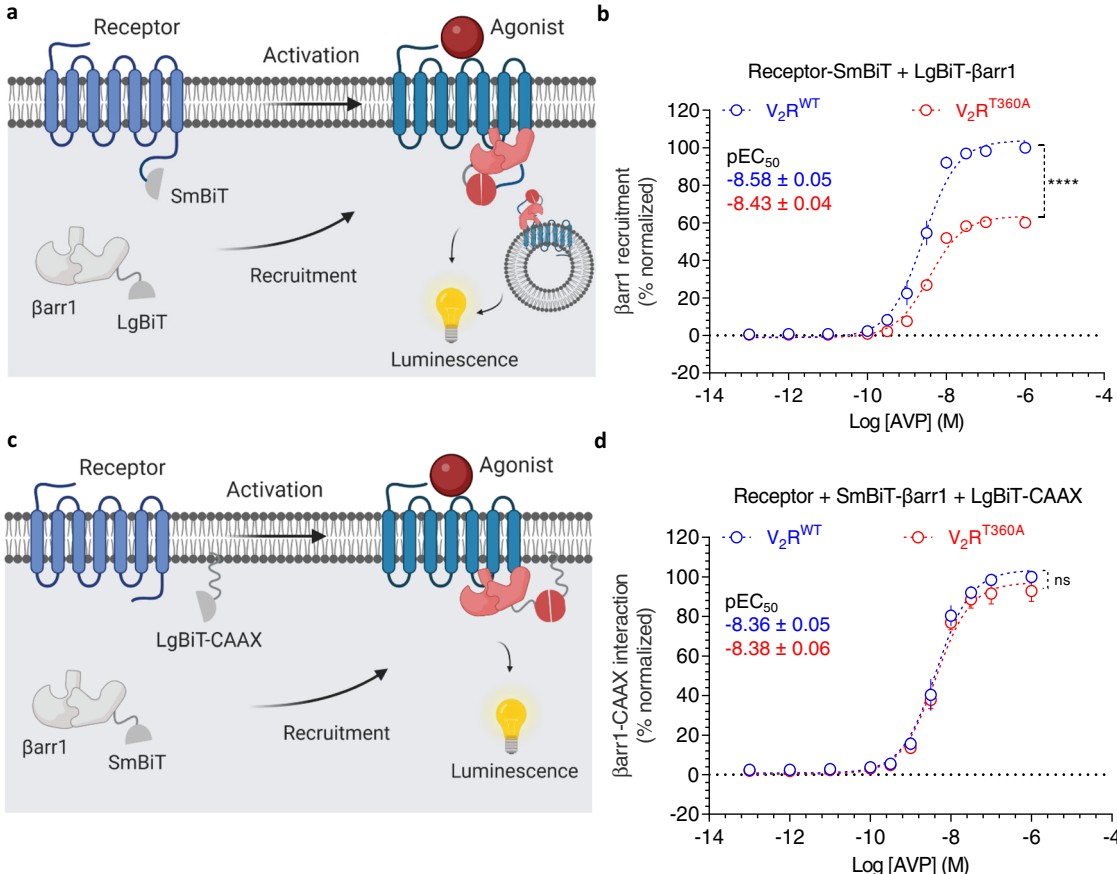

**Fig. 4 | Agonist-induced βarr1 recruitment to V₂R^WT and V₂R^T360A mutant.**
**a** Schematic representation of NanoBiT-based βarr1 recruitment assay. **b** HEK-293 cells expressing the indicated receptor and βarr1 constructs were stimulated with varying doses of AVP for 30 min followed by the measurement of luminescence (mean ± SEM; $n$ = 4 independent experiments; normalized with luminescence signal for V₂R^WT at maximal ligand dose as 100%, Two-way ANOVA, Sidak's multiple comparisons test; ****$p$ < 0.0001). **c** Schematic representation of NanoBiT-based

assay for measuring βarr1 translocation to the cell surface. **d** HEK-293 cells expressing the indicated receptor and βarr1 constructs together with LgBiT-CAAX were stimulated with varying doses of AVP for 30 min followed by the measurement of luminescence (mean ± SEM; $n$ = 4 independent experiments; normalized with luminescence signal for V₂R^WT at maximal ligand dose as 100%, Two-way ANOVA, Sidak's multiple comparison test; ns = non-significant). Source data are provided as a Source Data file.

previous study[13], it does not significantly potentiate ERK1/2 activation (Fig. 7a, b). As we are measuring ERK1/2 activation after five minutes of agonist-stimulation, where the signal is typically maximal and saturated, Ib30-mediated potentiation of ERK1/2 response for V₂R^WT, if any, may not be apparent under these conditions.

While previous studies have used intrabodies as biosensors of GPCR activation[23], βarr trafficking[13], inhibitors of GPCR endocytosis[24] and Gβγ signaling[25], the current study provides an example of an intrabody-based approach to positively modulate βarr trafficking and functional outcomes. Considering the earlier studies reporting sustained cAMP generation from the internalized pool of V₂R, we anticipated an enhanced cAMP response for the V₂R^T360A in presence of Ib30 due to potentiation of βarr1 endosomal trafficking. However, we did not observe a significant difference in cAMP response between Ib-CTL vs. Ib30 conditions for V₂R^T360A (Supplementary Fig. 6a). While this may simply reflect an inherent limitation of the GloSensor assay due to robust signal amplification, it would be interesting to investigate this aspect further in future studies. For example, in case of wild-type V₂R, agonist-stimulation promotes co-localization of the receptor, βarr1 and Ib30 in endosomal vesicles[13] however, it is plausible that V₂R^T360A dissociates from βarr1 at the plasma membrane due to relatively lower affinity. This may lead to trafficking of βarr1, presumably stabilized in an active conformation by Ib30, to endosomal vesicles even in the absence of the receptor. Further investigation along these lines in future studies may also help clarify the underlying mechanism for the

lack of cAMP potentiation in case of V₂R^T360A despite enhanced endosomal trafficking of βarr1 in the presence of Ib30.

An elegant study has recently reported crystal structures of βarr1 in complex with several different phospho-peptides derived from the carboxyl-terminus of V₂R, including the V₂Rpp^T360-1 [26]. While the binding affinities of βarr1 to V₂Rpp^WT and V₂Rpp^T360-1 are comparable, V₂Rpp^T360-1 exhibits a slightly altered binding mode compared to V₂Rpp^WT in these crystal structures[26]. Therefore, it is unlikely that distinct trafficking patterns of βarr1 for V₂R^WT vs. V₂R^T360A originate from an affinity difference, and it points towards a conformational mechanism underlying this phenomenon. This is indeed supported by our MD simulation studies, where Fab30 binding enriches active-like conformational populations in βarr1 with inter-domain rotation as a readout for the mutant phospho-peptide. GPCR-βarr interactions are typically thought to be biphasic and involve the phosphorylated carboxyl-terminus of the receptor and the cytoplasmic face of the activated transmembrane bundle[6,7,27–30]. Previous studies have visualized such partially-engaged and fully-engaged GPCR-βarr complexes and deciphered functional outcomes associated with these distinct conformations[28–30]. A recent study using NMR spectroscopy demonstrated that Fab30 binding to a partially-engaged GPCR-βarr complex facilitates additional conformational changes in βarr1 leading to a transition towards fully-activated conformation[31]. Our study now draws an interesting parallel with this recent NMR study by demonstrating that Ib30 allosterically facilitates the transition of a

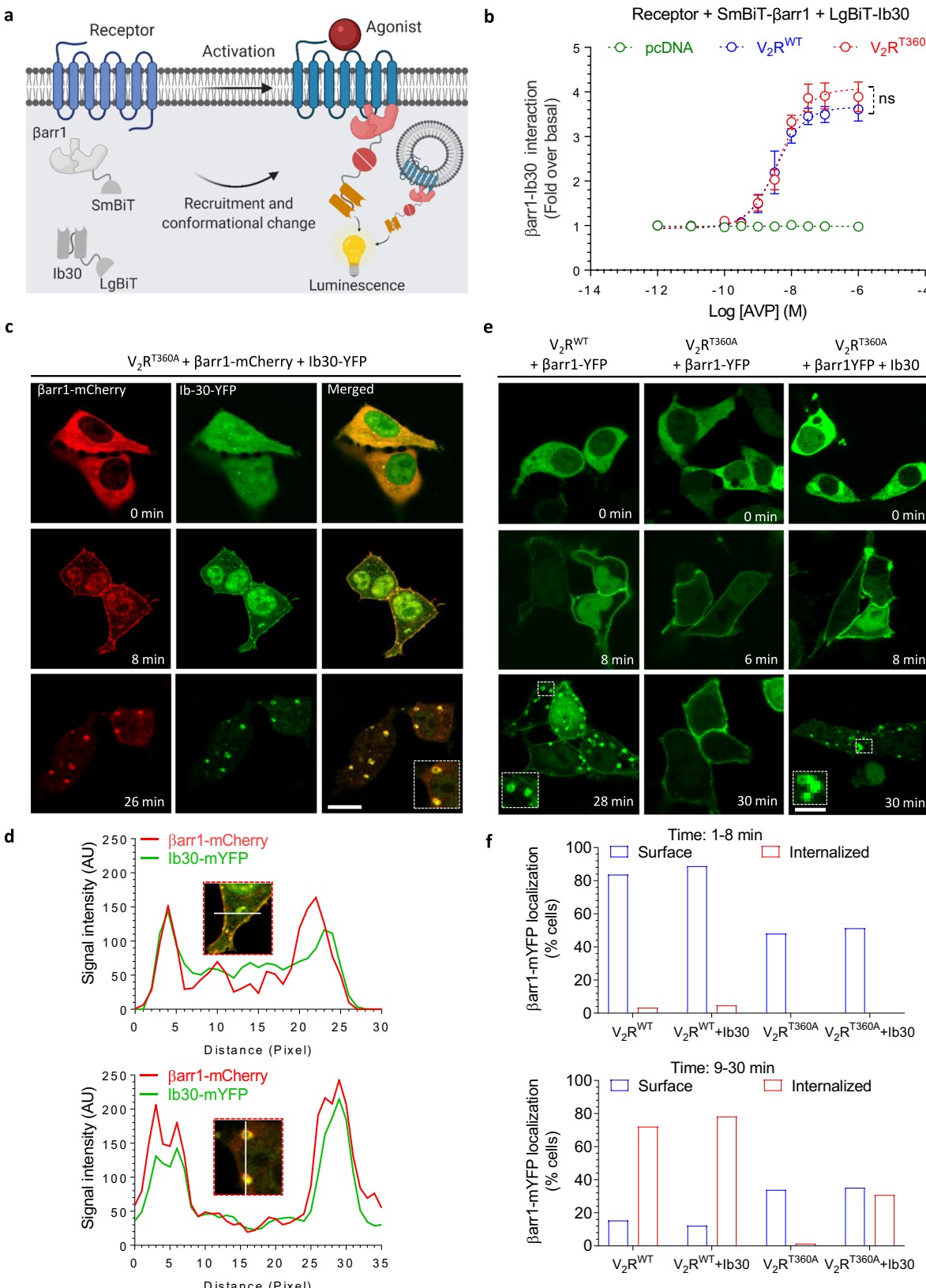

functionally-compromised βarr1 conformation to a fully-competent conformation for V₂R^T360A, and rescues downstream functional responses. It would be interesting to explore in future studies whether the effect of Ib30 observed here for V₂R^T360A is somehow linked to the transition between the partially- and fully-engaged βarr conformations in complex with the receptor.

The paradigm of βarr-AP2 interaction through β₂-adaptin in driving GPCR endocytosis via a clathrin-mediated pathway is mostly conserved across GPCRs[5]. Therefore, our study raises the possibility of Ib30 being a potentially generic positive modulator of βarr1 trafficking for other GPCRs as well, especially those exhibiting Class A pattern of βarr recruitment. Interestingly, we have already demonstrated the

**Fig. 5 | Intrabody30 sensor efficiently recognizes βarr1 conformation induced by $V_2R^{T360A}$. a** Schematic representation of NanoBiT-based assay to measure Ib30 reactivity to βarr1. **b** Recognition of βarr1 by Ib30 upon stimulation of $V_2R^{WT}$ or $V_2R^{T360A}$. HEK-293 cells expressing the indicated receptor, βarr1 and Ib30 constructs were stimulated with varying doses of AVP for 30 min, followed by the measurement of luminescence (mean ± SEM; $n = 4$ independent experiments; normalized with luminescence signal for $V_2R^{WT}$ at maximal ligand dose as 100%; Two-way ANOVA, Tukey's multiple comparisons test; ns = non-significant). **c** βarr1 and Ib30 are colocalized upon agonist-stimulation of $V_2R^{T360A}$. HEK-293 cells expressing $V_2R^{T360A}$ together with βarr1-mCherry and Ib30-YFP were stimulated with AVP (100 nM) for indicated time-points followed by visualization using confocal microscopy. Micrographs are representative of three independent experiments (Scale bar 10 μm). Uncropped micrographs are available in the Source Data file.

**d** Line-scan analysis of the indicated regions from confocal micrographs confirms the colocalization of βarr1 and Ib30. **e** Expression of Ib30 drives endosomal localization of βarr1 for $V_2R^{T360A}$. HEK-293 cells expressing $V_2R^{WT}$ or $V_2R^{T360A}$ together with βarr1-YFP were stimulated with AVP (100 nM), and the localization of βarr1 was monitored using confocal microscopy (Scale bar 10 μm). **f** The effect of Ib30 on localization of βarr1 as assessed by manually scoring HEK-293 cells from multiple fields in three independent experiments. Captured confocal images were grouped in two classes i.e., 1–8 min and 9–30 min post-agonist stimulation to monitor membrane and endosomal localization, respectively. The bar graphs indicate the % of cells showing βarr localization at the surface or in endosomal punctate structures in more than 500 cells for each condition collected from different field of views of three independent transfections and imaging experiments. Source data are provided as a Source Data file.

ability of Ib30 to recognize βarr1 in complex with several native GPCRs, although it was selected from a phage display library using $V_2$Rpp-βarr1 as the target[13,27]. Therefore, it would be worth probing the effect of Ib30 in the context of endocytosis and ERK1/2 phosphorylation for other receptors in future studies. Another interesting avenue where Ib30 may serve as a useful tool is the emerging paradigm of catalytic activation of βarrs, where they may continue to generate functional outputs even after dissociation from activated receptors[32–34]. It is plausible that Ib30 may recognize and stabilize such conformational "memory" in βarrs and thereby, facilitate its visualization in the cellular context as well as at high resolution using direct structural approaches.

In summary, we demonstrate that agonist-induced trafficking of βarrs and downstream responses can be allosterically modulated using conformation-specific intrabodies targeting protein-protein interactions. These findings open a paradigm for positively modulating GPCR signaling in cellular context and may catalyze the discovery of previously unknown aspects of GPCR-βarr interaction and functional outcomes.

## Methods

### General reagents

Most chemicals and molecular biology reagents were purchased from Sigma-Aldrich unless mentioned otherwise. HEK-293 cells (ATCC; cat. no. CRL-3216) were maintained at 37 °C under 5% $CO_2$ in Dulbecco's modified Eagle's medium (Gibco; cat. no. 12800-017) supplemented with 10% FBS (Gibco; cat. no. 10270-106) and 100 U ml$^{-1}$ penicillin and 100 μg ml$^{-1}$ streptomycin (Gibco; cat. no. 15140-122). Cells were cultured in 10 cm dishes (Corning; cat. no. 430167) at 37 °C under 5% $CO_2$ and passaged at 70 to 80% confluency using 0.05% trypsin-EDTA for detachment. Sf9 cells (Expression Systems; cat. no. 94-001 F) were maintained as suspension cultures in ESF 921 media (Expression Systems; cat. no. 96-001-01). Lauryl Maltose Neopentyl Glycol (LMNG) was purchased from Anatrace (cat. no. NG310).

### Construct design and expression plasmids

The expression constructs for the wild-type human $V_2R$ and $V_2R^{T360A}$ mutants have been described previously[10]. Briefly, the cDNA coding for $V_2R^{WT}$ with an N-terminal HA signal sequence and FLAG tag was PCR amplified and cloned in a customized pcDNA3.1 (+) vector. This construct was also cloned in pVL1393 vector for expression in Sf9 cells. The Thr$^{360}$ mutation was generated on the $V_2R^{WT}$ backbone using Q5 Site-Directed Mutagenesis Kit (NEB). The βarr1-mYFP plasmid used for confocal imaging experiments was obtained from Addgene (cat. no. 36916). βarr1-mCherry plasmid was a gift from Dr. Mark Scott, Institut Cochin, France. The plasmids encoding ScFv-CTL, ScFv30, Ib-CTL-HA, Ib30-HA and Ib30-YFP have been described previously[13,24]. The $V_2R^{WT}$ and $V_2R^{T360A}$ constructs were also fused with a 15 amino-acid flexible linker to the small subunit of NanoLuc i.e., SmBiT at its C-terminus. Similarly, Ib30 were N-terminally fused with LgBiT fragment in pCAGGS vector for NanoLuc complementation-based NanoBit assay. For in-vitro assays, i.e., trypsin proteolysis and ScFv30/Fab30 co-IP

experiments, βarr1 was purified from BL21 cells by Glutathione Sepharose (GS) affinity chromatography. All the constructs were sequence verified (Macrogen). $V_2R$ agonist AVP (arginine-vasopressin) was synthesized by Genscript, and phospho-peptides $V_2$Rpp$^{WT}$, $V_2$Rpp$^{T360-1}$, and $V_2$Rpp$^{T360-2}$ were synthesized by the peptide synthesis facility at Tufts University. The construct for GST-tagged β2-adaptin (residues 592–951, Rat, isoform 2) in pGEX4T1 vector was received as a kind gift from Dr. Thomas Pucadyil (Pune, India).

### Limited trypsin proteolysis assay

To qualitatively assess the effect of different $V_2R$ phospho-peptides i.e., $V_2$Rpp$^{WT}$, $V_2$Rpp$^{T360-1}$ and $V_2$Rpp$^{T360-2}$ on βarr1 conformation, limited trypsin proteolysis of βarr1 in the presence or absence of these phospho-peptides was performed. The protocol for trypsin proteolysis of βarr1 has been described previously[15]. Briefly, βarr1 (5–10 μM) was incubated in the absence or presence of (50:1 molar ratio, phospho-peptide: βarr1) the phospho-peptides for 30 min at 4 °C. Thereafter, L-1-Tosylamido-2-phenylethyl chloromethyl ketone (TPCK) treated Trypsin (Sigma-Aldrich; cat. no. T1426) was added to the βarr1 phospho-peptide mixture at a ratio of 1: 25 and 1: 50 (w/w) and the samples were incubated at 37 °C for 5 min. In addition to the indicated ratio of trypsin: βarr1, other ratios like 1: 10, 1: 100 and 1: 250 were also tried. At 1: 10 ratio, βarr1 was completely digested while at lower trypsin concentrations, the resolution of the digested fragments was poor. At each time point, 20 μl of the reaction mix (5 μg of βarr1) was withdrawn and transferred to a fresh microcentrifuge tube containing 5 μl of 5x SDS loading buffer to quench the proteolysis reaction. The digested samples were separated on 12% SDS-polyacrylamide gels by electrophoresis to determine the effect of phospho-peptides on the digestion pattern of βarr1. In addition, to study how ScFv30 affects the digestion pattern of βarr1 when activated with different phospho-peptides, a 50-fold molar excess of ScFv30 was added to the βarr1 samples prior to proteolysis. Samples without ScFv30 were used as references for comparison. After proteolysis with a 1: 50 ratio of trypsin: βarr1, the samples were quenched at 30 min and resolved by SDS-PAGE as described earlier.

### Surface expression of receptor mutants

The surface expression of $V_2R^{WT}$ and $V_2R^{T360A}$ used in different cellular assays was measured by whole-cell surface ELISA. For this, HEK-293 cells transfected with either $V_2R^{WT}$ or $V_2R^{T360A}$ were seeded at a density of 0.2 million per well in a 24-well plate precoated with 0.01% poly-D-Lysine (Sigma-Aldrich; cat. no. P0899). After 24 h, cells were fixed with 4% (w/v) paraformaldehyde (pH 6.9) on ice for 20 min and washed three times with 1× tris-buffered saline (TBS) buffer [150 mM NaCl and 50 mM Tris-HCl (pH 7.4)]. Subsequently, nonspecific sites were blocked with 1% bovine serum albumin (BSA; prepared in 1× TBS) for 90 min, followed by the incubation of cells with horseradish peroxidase (HRP)-coupled anti-FLAG M2 antibody (dilution-1: 5000; Sigma-Aldrich; cat. no. A8592), prepared in 1% BSA for 90 min. Cells were then washed three times with 1% BSA in TBS, and 200 μl of

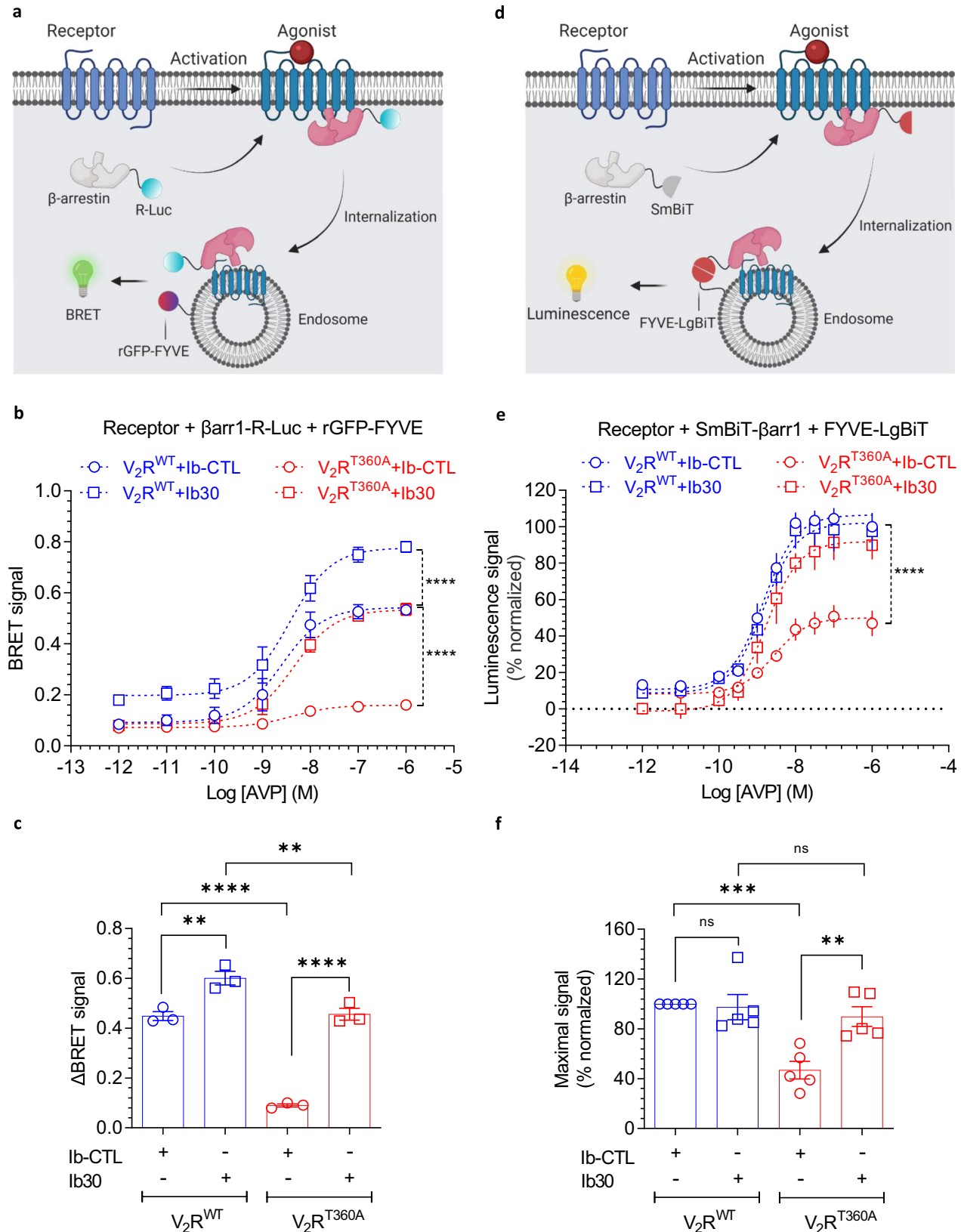

**Nature Communications** | (2022)13:4634

tetramethylbenzidine (TMB) ELISA substrate (Thermo Fisher Scientific; cat. no. 34028) was added to each well. Once blue color appeared in the wells, the reaction was stopped by transferring 100 μl of the solution to a different 96-well plate already containing 100 μl of 1 M $H_2SO_4$. Absorbance was measured at 450 nm in a multimode plate reader (Victor X4-Perkin-Elmer). For normalization of signal across

different wells, cell density was estimated using Janus Green (Sigma-Aldrich; cat. no. 201677) staining. TMB solution was removed from the wells; cells were washed with 1× TBS followed by incubation with 0.2% (w/v) Janus Green for 20 min. Thereafter, cells were washed three times with distilled water and 800 μl of 0.5 N HCl was added to each well. 200 μl of this solution was used for measuring the absorbance at

**Fig. 6 | Ib30 potentiates endosomal trafficking of βarr1 for the V₂R^T360A mutant.**
**a** Schematic representation of BRET-based endosomal localization assay for βarr1.
**b** Co-expression of Ib30 promotes endosomal trafficking of βarr1 for V₂R^T360A as assessed by BRET assay. HEK-293 cells expressing the indicated constructs were stimulated with varying doses of AVP followed by BRET measurement. Data (mean ± SEM) from three independent experiments are presented here (Two-way ANOVA, Tukey's multiple comparisons test; ****$p < 0.0001$). **c** Comparison of ΔBRET (difference in the BRET signal at the lowest and highest dose of AVP) in the BRET assay based on the data presented in panel B from three independent experiments (mean ± SEM, One-way ANOVA, Tukey's multiple comparisons test; **$p < 0.01$, ****$p < 0.0001$). **d** Schematic representation of NanoBiT-based

endosomal localization assay for βarr1. **e** Co-expression of Ib30 robustly promotes endosomal trafficking of βarr1 for V₂R^T360A as assessed by NanoBiT assay. HEK-293 cells expressing the V₂R^WT or V₂R^T360A, together with SmBiT-tagged βarr1 and Ib-CTL/Ib30 were stimulated with indicated doses of AVP followed by luminescence measurement. Data (mean ± SEM) from five independent experiments, normalized with maximal response under V₂R^WT + Ib-CTL condition (treated as 100%) (Two-way ANOVA, Tukey's multiple comparisons test; ****$p < 0.0001$) are presented here. **f** Comparison of maximal response (at 1 μM AVP) in the NanoBiT-based endosomal trafficking assay presented in panel E from five independent experiments (mean ± SEM, One-way ANOVA, Tukey's multiple comparisons test; **$p < 0.01$, ***$p < 0.001$, ns = non-significant). Source data are provided as a Source Data file.

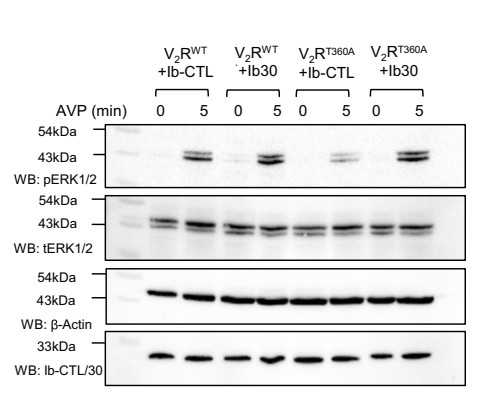

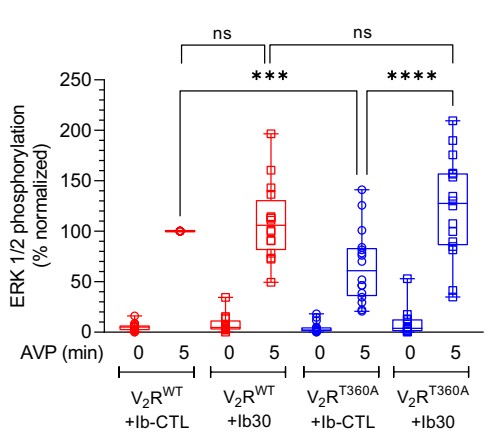

**Fig. 7 | Ib30 potentiates agonist-induced ERK1/2 activation for the V₂R^T360A mutant. a** HEK-293 cells expressing the indicated receptor construct together with Ib-CTL/Ib30 were stimulated with AVP (100 nM) followed by detection of ERK1/2 phosphorylation using western blot. The expression of Ib-CTL/Ib30 is monitored using anti-HA antibody. **b** Densitometry-based quantification from sixteen independent experiments, normalized with respect to V₂R^WT + Ib-CTL condition

(treated as 100%) (One-way ANOVA, Sidak's multiple comparisons test; ***$p = 0.0006$, ****$p < 0.0001$, ns = non-significant). The data represented as box plots showing median, IQR with whiskers of 1.5× IQR, and circles represent values from sixteen independent experimental replicates. Source data are provided as a Source Data file.

595 nm. Normalized surface expression of receptor constructs was calculated as the ratio of absorbance at 450 nm and 595 nm.

## NanoBiT assay

NanoBiT assays were carried out following a previously published protocol[35]. Briefly, HEK-293 cells were transfected with the plasmids as indicated in the corresponding figures using PEI (Poly-ethylenimine; 1 mg ml⁻¹) as transfection agent at a DNA: PEI ratio of 1: 3. For total βarr1 recruitment (Fig. 4b), 4 μg of receptor-SmBiT and 3 μg of LgBiT-βarr1 were used, while for measuring surface recruitment (Fig. 4d), 3 μg of receptor, 2 μg of SmBiT-βarr1 and 5ug of LgBiT-CAAX were transfected. For the Ib30 reactivity assay (Fig. 5b), 5 μg of receptor, 5 μg of LgBiT-Ib30, and 2 μg of SmBiT-βarr1 were transfected. For endosomal trafficking experiment (Fig. 6e), 3 μg of receptor, 2 μg of SmBiT-βarr1, and 5 μg of FYVE-LgBiT were transfected. To measure the effect of Ib30 on total βarr1 recruitment (Supplementary Fig. 5b), 4 μg of receptor-SmBiT, 3 μg of LgBiT-βarr1 and either 6 μg of Ib-CTL or 1 μg of Ib30 were used. After 16–18 h of transfection, cells were harvested in PBS solution containing 0.5 mM EDTA and centrifuged. Cells were resuspended in 3 ml assay buffer (HBSS buffer with 0.01% BSA and 5 mM HEPES, pH 7.4) containing 10 μM coelenterazine (GoldBio; cat. no. CZ05) at final concentration. The cells were then seeded in a white, clear-bottom, 96-well plate at a density of 0.7–0.9 × 10⁵ cells per 100 μl per well. The plate was kept at 37 °C for 90 min in the CO₂ incubator followed by incubation at room temperature for 30 min. Basal readings were taken in luminescence mode of a multi-plate reader (Victor X4-Perkin-Elmer). The cells were then stimulated with varying doses of ligand AVP ranging from 1 pM

to 1 μM (6x stock, 20 μl per well) prepared in drug buffer (HBSS buffer with 5 mM HEPES, pH 7.4). Luminescence was recorded for 60 min immediately after addition of ligand. The initial counts of 4–10 cycles were averaged and fold increase was calculated with respect to vehicle control (unstimulated values) and analyzed using nonlinear regression four-parameter sigmoidal concentration–response curve in GraphPad Prism software (v9.3).

## Confocal microscopy

For visualizing the effect of intrabody on βarr-mediated receptor trafficking, HEK-293 cells were co-transfected with 3 μg of either V₂R^WT or V₂R^T360A along with 2 μg of βarr1-mYFP in the presence or absence of 2 μg of Ib30 with help of polyethylenimine (Polysciences; cat. no. 23966) reagent (21 μl) in 10 cm plates. Transfection was performed in FBS-deficient DMEM (Gibco; cat. no. 12800-017) after which cells were replaced with DMEM supplemented with FBS (Gibco; cat. no. 10270-106). Post 24 h, cells were seeded onto poly-D-lysine (Sigma-Aldrich; cat. no. P0899) precoated glass bottom confocal dishes (SPL Life-sciences; cat. no. 100350) at a density of 1 million per dish. Cells were allowed to adhere to confocal dishes for 24 h. The next day, cells were starved in FBS-deficient DMEM for 4 h and then stimulated with 100 nM AVP, and live cells were visualized under the confocal micro-scope (Zeiss LSM 710 NLO). The confocal microscope was equipped with a motorized XY stage along with a temperature and CO₂ con-trolled platform. For visualizing Ib30 and βarr1 together, cells were transfected with βarr1-mCherry (2 μg) and Ib30-mYFP (2 μg) along with V₂R^T360A (3 μg). To excite mYFP, a multi-line argon laser source was used and for the mCherry, a diode pump solid-state laser source was used.

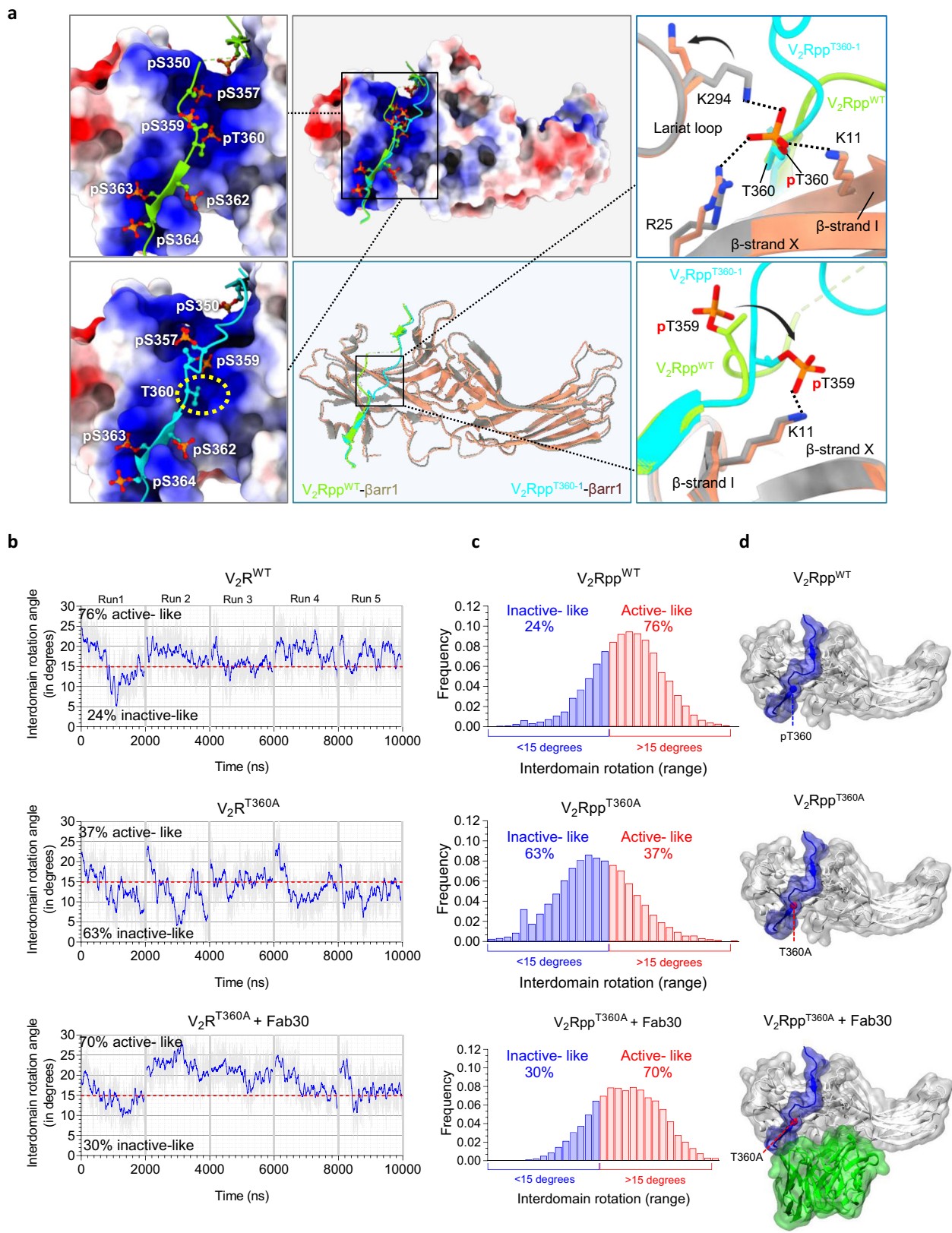

The emitted signal was detected with a 32× array GaAsP descanned detector (Zeiss). For related experiments, all microscopic settings including laser intensity and pinhole slit were kept in the same range and for avoiding any spectral overlap between two channels filter excitation regions and bandwidths were adjusted accordingly. Images were acquired in line scan mode and were subsequently processed

post imaging in ZEN lite (ZEISS) software suite. For quantifying βarr trafficking to either membrane or endosomes, confocal images were categorized into early (1 to 8 min) and late time points (9 to 30 min) post agonist stimulation. The cells with βarr1-mYFP fluorescence in the plasma membrane were scored as surface localized, and the cells with punctate structures in the cytoplasm were scored as internalized. In

**Fig. 8 | Intrabody30 (Ib30) stabilizes the active conformation of βarr1.**
**a** Structural snapshots of βarr1 crystal structures in complex with $V_2Rpp^{WT}$ (PDB: 4JQI) and $V_2Rpp^{T360-1}$ (PDB: 7DFA). The superimposed structures display repositioning of the $V_2Rpp^{T360-1}$ N-terminal segment harboring $Thr^{360}$ residue (cyan) relative to the $V_2R^{WT}$ (green). Also, changes in ionic interactions of $Thr^{360}$ with neighboring residues are shown. For the $V_2R^{WT}$ bound βarr1, $Thr^{360}$ engages with $Lys^{294}$, $Lys^{11}$, and $Arg^{25}$. In the $V_2R^{T360-1}$ bound state, the $Thr^{360}$ is non-phosphorylated, and the side-chain of $Thr^{359}$ is repositioned to interact with $Lys^{11}$. **b–d** MD

simulation of βarr1 in complex with either $V_2Rpp^{WT}$ or $V_2Rpp^{T360A}$ based on the crystal structure of $V_2Rpp$-βarr1 (PDB: 4JQI) reveals enrichment of inactive-like conformations of βarr1 in $V_2Rpp^{T360A}$-bound conformation. However, the binding of Fab30 to $V_2Rpp^{T360A}$-βarr1 complex robustly enriches the active-like conformational population of βarr1 as assessed by inter-domain rotation. The blue line in **8b** represents the rolling averages, while the grey line represents the original values for interdomain rotation per frame. Source data are provided as a Source Data file.

cases where βarrs were seen in both the membrane and in cytoplasmic punctate structures, cells having more than three punctae in the cytoplasm were scored under internalized category. Biological replicates were imaged at least three times independently on different days. Scored data from the cell count were plotted as the percentage of βarr recruitment from more than 500 cells for each condition. To avoid any discrepancies in manual counting, three different individuals counted the images in a blinded and cross-checked fashion. All data were plotted in GraphPad Prism software (v9.3).

### Agonist-induced cAMP responses measured by GloSensor assay

To measure cAMP accumulation (as a readout for G protein activation), 50–60% confluent HEK-293 cells were co-transfected with either $V_2R^{WT}$ or $V_2R^{T360A}$ DNA (2 μg), luciferase-based 22 F cAMP biosensor construct (3.5 μg) and Ib-CTL (2 μg) or Ib30 (1 μg) DNA. After 18–20 h of transfection, cells were washed with 1xPBS and treated with trypsin-EDTA (0.05%). Detached cells were harvested and centrifuged at 184 X g for 10 min, and the cell pellet was resuspended in 0.5 mg ml$^{-1}$ luciferin (GoldBio; cat. no. LUCNA) solution prepared in 1X HBSS buffer (Gibco; cat. no. 14065) containing 20 mM HEPES (pH 7.4). Cells were then seeded at a density of 0.1–0.125 million per 100 μl in 96 well white plate. The same pool of cells was also seeded side by side for surface expression by whole cell surface ELISA. The cells seeded in 96-well plate were incubated for 1.5 h in 5% $CO_2$ followed by additional 30 min at room temperature. Subsequently, the basal luminescence was recorded for 5 cycles using a plate reader (Victor X4-Perkin-Elmer), followed by the addition of indicated concentrations of agonist AVP and luminescence was recorded for 1 h (30 cycles). Data were corrected for baseline signal and percent normalized with respect to maximal agonist concentration of $V_2R^{WT}$ + Ib-CTL.

### BRET assay for βarr1 trafficking

HEK-293T cells (ATCC) were grown in complete culture media (DMEM high glucose (Wisent; cat. no. 319-015-CL) supplemented with 10% FBS (Wisent; cat. no. 098150) and penicillin/streptomycin (Wisent; cat. no. 450-201-EL) in a tissue culture incubator set at 37 °C providing 5% $CO_2$. The day before transfection, cells were plated into well of a 6-well plate (Thermo scientific; cat. no. 140675) at 400,000 cells per well. The next day, media was changed for DMEM high glucose supplemented with only 2.5% FBS and cells were transfected using PEI (Polysciences; cat. no. 23966) as follows: 1 μg total DNA composed of 10 ng of $V_2R^{WT}$ or $V_2R^{T360A}$, 25 ng of βarr1-RlucII, 100 ng of rGFP-FYVE and either 300 ng Ib-CTL or 50 ng Ib30 and DNA amount was completed with pcDNA3.1(+) was mixed with 3 μl of a 1 mg ml$^{-1}$ PEI solution and added drop-wise to cells. The plate was put back in the incubator till the next day. 24 h post-transfection, cells were detached and re-plated at 60,000 cells per well into a poly-L-ornithine-coated (Sigma-Aldrich; cat. no. P3655) white 96-well plate (Thermo scientific; cat. no. 236105) in complete culture media and left to grow for another 24 h. Then, the 96-well plate was washed once with Kreb's/HEPES solution (146 mM NaCl, 4.2 mM KCl, 0.5 mM $MgCl_2$, 1 mM $CaCl_2$, 5.9 mM glucose, and 10 mM HEPES buffer, pH 7.4) and 80 μl of Kreb's/HEPES was added per well. The plate was put back in the incubator for 2–3 h to allow cells to rest before BRET measurement. After the resting time, cells were stimulated for 15 min at 37 °C by adding 10 μl of AVP (Sigma-Aldrich; cat. no. V9879) at different concentrations prepared in Kreb's/HEPES. To

assess BRET, 10 μl of a 20 μM coelenterazine 400 A (GoldBio; cat. no. C-320) solution diluted in Krebs/HEPES was added 5 min before the end of the stimulation period. BRET was then monitored by measuring 3 consecutive luminescence readings at both 410 nm and 515 nm using a Tristar2 plate reader (Berthold. Technologies GmbH & Co. KG). BRET was calculated as the emission at 515 nm/emission at 410 nm and the 3 values were averaged. BRET data were plotted as dose-response curves using GraphPad Prism (v6).

### Effect of Ib30 on agonist induced ERK1/2 phosphorylation

To assess the effect of Ib30 on βarr mediated signaling downstream to $V_2R^{WT}$ and $V_2R^{T360A}$ mutant, agonist-induced ERK1/2 phosphorylation was measured. For this, 60–70% confluent HEK-293 cells were co-transfected with 0.25 μg of indicated $V_2R$ constructs and 1 μg of HA-tagged Ib30. A control intrabody (Ib-CTL) that does not recognize receptor-bound βarr1 was also transfected in parallel at levels comparable to Ib30 (3 μg) to achieve normalized expression levels of both the intrabodies. 24 h after transfection, cells were seeded into six-well plates at a density of 1 million cells per well. The next day, cells were serum-starved in DMEM for 6 h and were then stimulated with 100 nM AVP (agonist for $V_2R$) for indicated time points. After stimulation for selected time points, the media was aspirated and the cells were lysed in 100 μl of 2× SDS protein loading buffer. Cellular lysates were heated at 95 °C for 15 min, followed by centrifugation at 21130 X g for 15 min. 10 μl of samples were loaded per well and separated by 12% SDS-polyacrylamide gel electrophoresis. Phosphorylated ERK1/2 signal was detected by Western blotting using anti–phospho-ERK1/2 antibody (dilution-1: 5000; CST; cat. no. 9101) followed by reprobing of the blots with anti–total-ERK1/2 antibody (dilution-1: 5000; CST; cat. no. 9102). Since the anti–phospho-ERK1/2 and anti–total-ERK1/2 antibodies were not coupled to HRP, the Anti-Rabbit IgG-Peroxidase antibody (dilution-1: 5000; Sigma-Aldrich; cat. no. A9169) was used for signal detection. The expression of Intrabody was confirmed by probing with anti-HA antibody (dilution-1: 5000; Santa-Cruz; cat. no. sc-805). β-actin expression is used as a loading control (dilution-1: 50000; Sigma, Cat. no. A3854). Signal on the western blots was detected using the ChemiDoc imaging system (Bio-Rad), and densitometry-based quantification was carried out using ImageJ software suite.

### Molecular dynamics simulations

Data without Fab30 was adapted from a previous study[10]. To generate $V_2Rpp^{WT}$-βarr1, $V_2Rpp^{360A}$-βarr1, and $V_2R^{T360A}$-βarr1-Fab30 complexes, we used previously determined crystal structure[27]. The sequence of βarr1 was reverted to match the isoform used in the in-vitro experiments [Uniprot AC: P29066]. The phosphorylation state of the $V_2Rpp$ was retained from the used crystal structures. Missing fragments in the βarr1 and $V_2Rpp$ structures were modeled using the loop modeller module available in the MOE package (www.chemcomp.com). In Fab30 we maintained residues 5 to 108 of the light chain and residues 1 to 123 of the heavy chain. The complexes were solvated (TIP3P water) and neutralized using a 0.15 concentration of NaCl ions. System parameters were obtained from the Charmm36M forcefield[36]. Simulations were carried out using the ACEMD3 engine[37]. Both systems underwent a 20 ns equilibration in conditions of constant pressure (NPT ensemble, pressure maintained with Berendsen barostat, 1.01325 bar pressure), using a timestep of 2 fs. During this stage

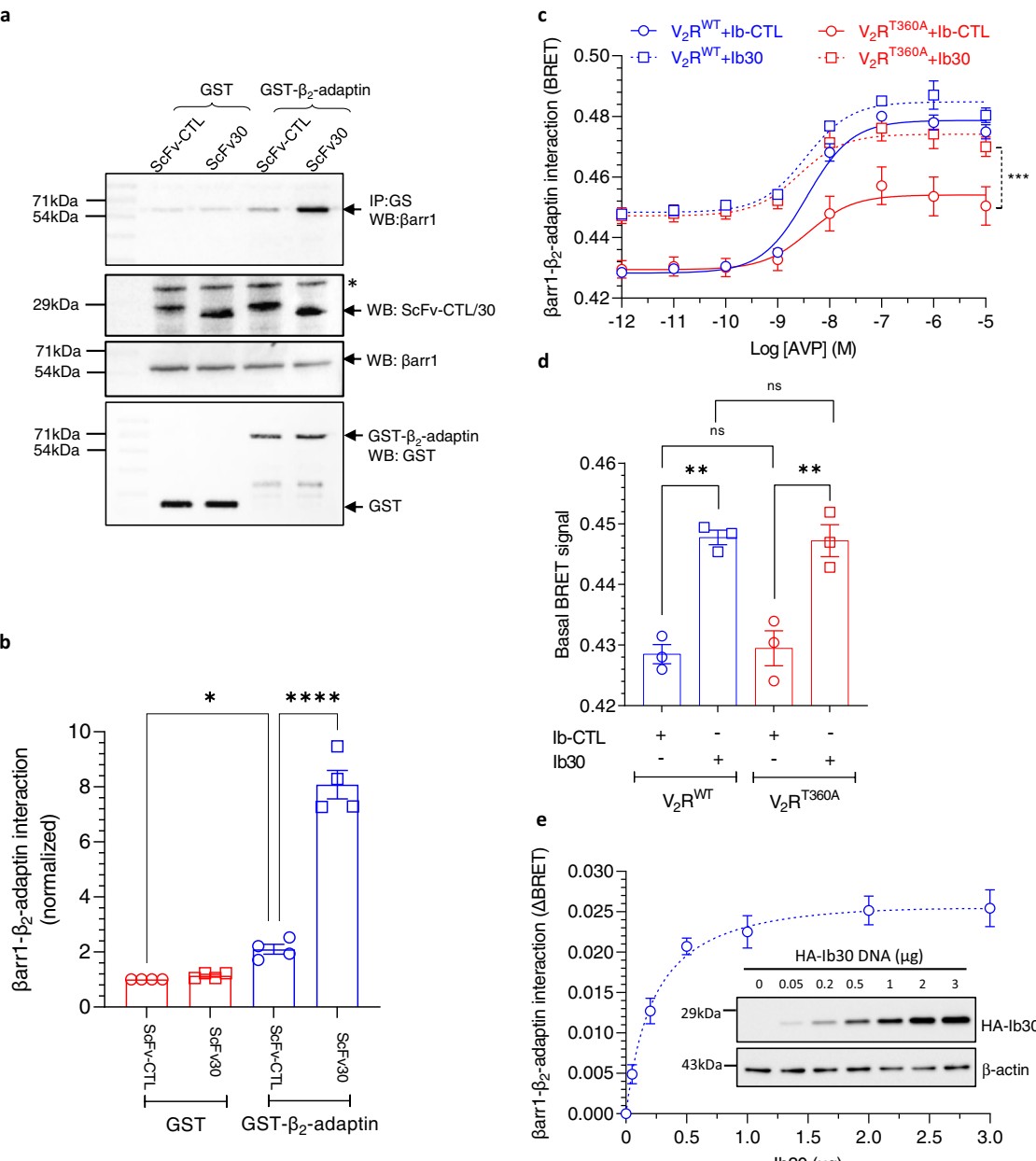

**Fig. 9 | Intrabody30 enhances the interaction of β₂-adaptin with βarr1.**
**a** Purified GST-β₂-adaptin (592–951) was incubated with $V_2R^{T360A}$ and βarr1 in presence of ScFv-CTL or SvFv30 followed by co-IP and Western blotting. Unconjugated GST was used as a negative control. A representative blot from three different experiments is shown here. The * symbol designates a non-specific band that we typically observe in lysates prepared from *Sf9* cells. **b** Densitometry-based quantification (mean ± SEM) of βarr1-β₂-adaptin interaction from four independent experiments normalized with GST control (One-way ANOVA, Sidak's multiple comparisons test; *p = 0.0308, ****p < 0.0001). **c** BRET between RlucII-tagged βarr1 and YFP-tagged β₂-adaptin shows enhanced interaction between βarr1 and β₂-adaptin in presence of Ib30, as compared to Ib-CTL, for both $V_2R^{WT}$ and $V_2R^{T360A}$. Data (mean ± SEM) from three independent experiments (Two-way ANOVA,

Tukey's multiple comparisons test; ***p = 0.0006) are presented here. **d** The BRET signal at lowest ligand concentration under different conditions as measured in panel C from three independent experiments (mean ± SEM; One-way ANOVA, Tukey's multiple comparisons test; $V_2R^{WT}$ (p = 0.0012); $V_2R^{T360A}$ (p = 0.0020), ns = non-significant). **e** Ib30 induced increase in βarr1-β₂-adaptin interaction exists even in the absence of either $V_2R^{WT}$ and $V_2R^{T360A}$. βarr1-β₂-adaptin interaction in presence of Ib30 exhibits a concentration-dependent increase until saturating concentration of the latter. Data represent four independent experiments (mean ± SEM). The *inset* shows a representative blot indicating the concentration range of Ib30 used in the BRET experiment, expression level and loading control (β-actin). Source data are provided as a Source Data file.

restraints were applied to the backbone. This was followed with 5 × 2 μs of simulation for each system in conditions of constant volume (NVT ensemble) using a timestep of 4 fs. This allowed us to amass a total of 10 μs simulation time per system. Simulations of inactive βarr1, as well as the $V_2Rpp^{T360-1}$ βarr1 complex were carried out in a 3 x 500 ns setup. For each of the simulations we used a temperature of 310 K, which was maintained using the Langevin thermostat, hydrogen bonds were

restrained using the RATTLE algorithm. Non-bonded interactions were cut-off at a distance of 9 Å, with a smooth switching function applied at 7.5 Å. The inter-domain rotation angle of βarr1 was analysed using a script kindly provided by Naomi Latoracca[38]. The angle was measured by comparing the displacement of the C-domain relative to the N-domain between the inactive (PDB code: 1G4R) and active βarr1 crystal structures (PDB code: 4JQI). Each simulation frame was aligned

to the reference structures using the Cα atoms of the β-strands present within the N-domain, while the same atoms present in the C-domain were used to calculate the rotation angle. We have deposited all the simulation data presented in the current manuscript in the GPCRmd portal.

## Co-immunoprecipitation (co-IP) assay

Co-IP was performed to evaluate the interaction between $V_2Rpp^{WT}$, $V_2RppT^{360-1}$ and $V_2Rpp^{T360-2}$ with βarr1 in presence of Fab30 and ScFv30. 5 µg of purified βarr1 was activated with 10-fold and 50-fold molar excess of phospho-peptides for 1 h at room temperature (25 °C) in binding buffer (20 mM HEPES, pH 7.4, 100 mM NaCl). Thereafter, the activated βarr1 was incubated with 2.5 µg of purified Fab30 or ScFv30. Subsequently, 20 µl of pre-equilibrated Protein L beads (GE Lifesciences; cat. no. 17547802) were added to the reaction mixture and incubated for an additional 1 h at room temperature, which was followed by extensive washing (3–5 times) with binding buffer + 0.01% LMNG. Elution was taken with 2X SDS loading buffer. Interaction of Fab30 and ScFv30 with βarr1 in presence of phospho-peptides was visualized using Coomassie staining of the gels. Band intensity was analysed by ImageJ gel analysis software.

To assess the effect of ScFv30 on $V_2R^{T360A}$ induced βarr1-$β_2$-adaptin interaction, we performed co-immunoprecipitation assay (co-IP). The $V_2R^{T360A}$ receptor was expressed in *Sf*9 cells, stimulated with 100 nM AVP and centrifuged to obtain receptor pellet. The receptor pellet was resuspended in appropriate volume of lysis buffer having 20 mM HEPES, 150 mM NaCl, 1X PhosSTOP (Roche; cat. no. 04906837001), and 1X protease inhibitor (Roche; cat. no. 04693116001), subjected to Dounce homogenization and incubated with 1 µg of purified βarr1 for 30 min at room temperature. The receptor-βarr complex was again incubated with 5 µg of purified ScFv30 or ScFv-CTL for another 30 min and solubilized with 1% LMNG for 1 h. Meanwhile, GST or GST-$β_2$-adaptin protein (2.5 µg) was immobilized on 20 µl buffer (20 mM HEPES, 150 mM NaCl) equilibrated GS beads (1 h at room temperature) and washed once to remove any unbound protein. Subsequently, the supernatant from solubilized complex was allowed to bind with protein bound GS beads (1 h at room temperature) followed by three washes with wash buffer (20 mM HEPES, 150 mM NaCl, 0.01% LMNG). The bead-bound complex was eluted in 2X SDS loading buffer. Eluted samples were separated by 12% SDS–polyacrylamide gel electrophoresis and probed using βarr antibody (dilution-1: 10000; CST; cat. no. 4674). After solubilization, 20 µl of lysate was set aside for confirming equal loading of βarr1 and ScFv. The lysate was run on separate 12% SDS–polyacrylamide gel and probed using βarr antibody and HRP-coupled protein L antibody (dilution-1: 2000; GenScript; cat. No. M00098) by western blotting. Band intensity was analysed by Image Lab software (Bio-Rad).

## BRET assay for βarr1-$β_2$-adaptin interaction

To monitor βarr1 and $β_2$-adaptin interactions, BRET assays between βarr1-RlucII and $β_2$-adaptin-YFP were performed as described[22]. HEK-293 cells were seeded at a density of $1 × 10^6$ cells per 100 mm dish and transfected the next day with 250 ng of $V_2R^{WT}$ or $V_2R^{T360A}$ along with 120 ng of βarr1-RlucII, 1 µg of $β_2$-adaptin-YFP, and either 1.5 µg Ib-CTL or 1 µg Ib30 using PEI. Briefly, a total of 6 µg of DNA (adjusted with pcDNA3.1/zeo(+)) in 0.5 ml of PBS was mixed with 12 µl of PEI (25 kDa linear, 1 mg ml⁻¹) in 0.5 ml PBS and then incubated for 20 min prior to applying to the cells. After 24 h, cells were detached and seeded onto poly-ornithine-coated 96-well white plates at a density of ~35,000 cells per well for the BRET assays, which were performed 48 h after transfection. For BRET assays, cells in 96-well plates were washed once with Tyrode's buffer (140 mM NaCl, 2.7 mM KCl, 1 mM $CaCl_2$, 12 mM $NaHCO_3$, 5.6 mM D-glucose, 0.5 mM $MgCl_2$, 0.37 mM $NaH_2PO_4$, 25 mM HEPES, pH 7.4) and left in Tyrode's buffer for 1 h at room temperature. Cells were stimulated with various concentrations of AVP for 45 min

then BRET signals were measured using a plate reader (Victor X4-Perkin-Elmer). Coelenterazine h (Nanolight™, final concentration of 5 µM) was added 25 min prior to BRET measurement. The filter set used was 460/80 nm and 535/30 nm for detecting the RlucII, *Renilla* luciferase (donor) and YFP (acceptor) light emissions, respectively. The BRET ratio was determined by calculating the ratio of light emitted by YFP over light emitted by RlucII. For Ib30 titration, HEK-293 cells were transfected with 120 ng of βarr1-RlucII and 1 µg of $β_2$-adaptin-YFP along with various amounts (0 to 3 µg) of Ib30 in 100 mm dishes or scaled down to 1/6 in a well in 6well plates. BRET signals were measured in absence of ligand stimulation. Expression levels of Ib30 were accessed by western blotting with anti-HA-peroxidase conjugate (dilution-1: 1000, Sigma-Aldrich; cat. no. 12013819001). Anti-β-actin antibody (dilution-1: 2000, Santa Cruz Biotechnology; cat. no. sc-47778) was used for loading control.

## Data quantification and statistical analysis

The experiments were conducted at least three times and data (mean ± SEM) were plotted and analyzed using GraphPad Prism software (v9.3). The data were normalized with respect to proper experimental controls and appropriate statistical analyses were performed as indicated in the corresponding figure legends.

## Reporting summary

Further information on research design is available in the Nature Research Reporting Summary linked to this article.

## Data availability

The original raw data for gels, immunoblots and confocal micrographs have been deposited in Mendeley Data (https://doi.org/10.17632/8wmkcw8ht7.1). This paper does not report any original code. The coordinates for $V_2Rpp^{WT}$-βarr1 and $V_2Rpp^{T360-1}$-βarr1 crystal structures used in this study are available in PDB with ID 4JQI and 7DFA, respectively. Any additional information required to reanalyze the data reported in this paper is available from the corresponding author upon reasonable request. Source data are provided with this paper. Original data pertaining to MD simulation are deposited in GPCRmd (https://submission.gpcrmd.org/dynadb/publications/1486/). Source data are provided with this paper.

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

## Acknowledgements

Research in A.K.S.'s laboratory is currently supported by the Senior Fellowship of the Wellcome Trust/DBT India Alliance (IA/S/20/1/504916) awarded to A.K.S., Department of Biotechnology (DBT) (BT/PR29041/BRB/10/1697/2018), Science and Engineering Research Board (EMR/2017/003804, SPR/2020/000408, and IPA/2020/000405), Council of Scientific and Industrial Research [37(1730)/19/EMR-II], Young Scientist Award from Lady Tata Memorial Trust, and IIT Kanpur. A.K.S. is an EMBO Young Investigator and Joy Gill Chair Professor. M.B. was supported by the National Post-Doctoral Fellowship of SERB (PDF/2016/002930) and Institute Post-Doctoral Fellowship of IIT Kanpur. H.D.-A. is supported by National Post-Doctoral Fellowship of SERB (PDF/2016/002893) and BioCare grant from DBT (BT/PR31791/BIC/101/1228/2019). M.C. is supported by a fellowship from CSIR [09/092(0976)/2017-EMR-I]. The work in T.E.H.'s laboratory was supported by a grant from the Canadian Institutes of Health Research (PJT-15698) and T.E.H. holds the Canadian Pacific Chair in Biotechnology. The work in S.A.L.'s laboratory was supported by the Canadian Institutes of Health Research: PJT-162368 and PJT-173504. J.S. is supported by the Instituto de Salud Carlos III FEDER (PI18/00094) and the ERA-NET NEURON & Ministry of Economy, Industry, and Competitiveness (AC18/00030). T.M.S. is supported by the National Science Centre of Poland, project number 2017/27/N/NZ2/0257. We thank Dr. Archana for help with some β2-adaptin co-IP experiments. A.I. was funded Japan by Society for the Promotion of Science (JSPS) KAKENHI grants 21H04791, 21H05113, JPJSBP120213501 and JPJSBP120218801; FOREST Program JPMJFR215T and JST Moonshot Research and Development Program JPMJMS2023 from Japan Science and Technology Agency (JST); The Uehara Memorial Foundation; and Daiichi Sankyo Foundation of Life Science.

## Author contributions

M.B. (MBA) carried out confocal microscopy, assisted in the NanoBiT assay, β2-adaptin interaction experiments using co-immunoprecipitation and ERK1/2 MAP kinase activation experiments; M.C. generated the receptor constructs with the help from P.S., carried out the limited proteolysis experiment with A.R., participated in β2-adaptin interaction experiments using co-IP, and performed ERK1/2 MAP kinase activation experiments with S.P.; HD-A carried out GloSensor assay with the help from M.C. and NanoBiT assay with MBA and PS, participated in β2-adaptin interaction experiments using co-IP; M.B. (MB), B.P., and M.K.Y. performed Fab30/ScFv30 co-IP assay; R.B. and J.M. carried out structural analysis of crystal structures; D.D. performed BRET experiments to monitor endosomal trafficking of βarr1 under the

supervision of TEH; T.M.S. performed MD simulation studies under the supervision of J.S.; Y.N. performed βarr1-β2-adaptin BRET experiments under the supervision of S.A.L.; K.K. and A.I. provided new reagents; all authors contributed in data interpretation and manuscript writing; T.E.H. and S.A.L. edited the manuscript; A.K.S. coordinated and supervised the overall project.

## Competing interests

The authors declare no competing interests.
