## [Peer Review File · Nature Communications]

Allosteric modulation of GPCR-induced β -arrestin trafficking and signaling by a synthetic intrabodyREVIEWER COMMENTS

Reviewer #1 (Remarks to the Author):

Key results:

In this study, the authors characterise the ability of intrabody30 (Ib30) to rescue β -arrestin1 translocation to endosomes impaired by a mutation of a phosphorylation site in the carboxy-terminal region of the vasopressin type 2 receptor (V_{2R}) previously reported by the same group. The authors demonstrate that Ib30 also rescues agonist-induced ERK1/2 phosphorylation by the mutant V_{2R} (V_{2R}^{T360A}), an event downstream and dependent of β -arrestin1 translocation to endosomes. Molecular dynamics simulations previously suggested that V_{2R}^{T360A} reduces the fraction of active β -arrestin1 conformation and in the present study elegant in vitro techniques were used to demonstrate that Ib30 rescues β -arrestin1 translocation to endosomes and agonist-induced ERK1/2 phosphorylation by enriching the active-like conformation population of β -arrestin1. The authors also demonstrate that Ib30 enhances the interaction between β -arrestin1 and β_{2} -adaptatin, which is a prominent mechanism that drives receptor internalisation. They hypothesise that Ib30-mediated increase of β -arrestin1/ β_{2} -adaptatin interaction is the possible mechanistic basis linking active β -arrestin1 enrichment and improved β -arrestin1 translocation to endosomes induced by Ib30. Overall these exciting findings provide a proof-of-concept that conformation-specific intrabodies represent a novel molecular tool to allosterically modulate agonist-induced trafficking of β -arrestins and therefore constitute an attractive solution to enhance this pathway in a therapeutic context.

Validity:

The interpretation and conclusions are based on solid data obtained from both in vitro and cell-based assays. The authors frequently use several complementary approaches to measure a given parameter, which strengthens the conclusions and considerably reduces the risk of conclusions based on artefacts.

Significance:

Endosomes represent a fantastic signalling platform where specific cellular functions can only occur from these intracellular compartments. For some receptors, ERK1/2 activation is an example, but since the past decade, a growing number of studies report a major functional role of G protein signalling from endosomes in cellular physiology. As Ib30 induces an enrichment of fully-active β -arrestins promoting their trafficking to endosomes, this intrabody could theoretically be used to positively modulate

downstream signalling dependent of the presence of β -arrestins in endosomes by any GPCR of interest (not only V_{2R}). Supporting this idea, the authors report that Ib30 efficiently recognises β -arrestin1 in complex with several native GPCRs (ref. 12, 22). Some studies report that β -arrestin binding to class B receptors in endosomes potentiates endosomal G protein activation by these receptors. Therefore, by promoting endosomal trafficking of β -arrestin, Ib30 could represent an interesting tool to increase endosomal G protein signalling of these receptors, and consequently selectively potentiate their downstream cellular functions. Additionally, by its ability to recognise the active conformation of β -arrestin, Ib30 could potentially be utilised in the context of drug discovery to screen compounds stabilising a receptor conformation promoting endosomal β -arrestin signalling. Overall, the present study will certainly be of great significance to the field of G protein-coupled receptor (GPCR) signalling from fundamental research to investigate the role of endosomal β -arrestin signalling to applied research to design new therapeutics targeting cellular functions downstream of β -arrestin signalling from endosomes.

Data and methodology:

Enough details are provided in the methods for the work to be reproduced. The methodology is overall elegant and adapted to the questions investigated. The authors combine a well-balanced ensemble of in vitro techniques, cell-based assays and computational approaches to answer their questions. In several situations, the authors combine different approaches together to answer a given question. The work certainly meet the expected state-of-the-art standards in the field.

Analytic approach:

The analytical approach is appropriate. However, in Fig. 1D, as data are normalized to V_{2Rpp}^{WT} , the % band protection to this condition is always 100. Consequently, the variance between the 4 experimental groups is not homogenous and therefore an ANOVA cannot be performed. In that situation, a one-sample t test and comparison to 100 would be the proper statistical test. Same thing for Fig. 2B, 4D, 6B, S1B, S2B, S3A, and S4A.

Suggested improvements:

The overall quality of the manuscript is excellent. I do not recommend any additional experiment. However, I have the following minor concerns to improve even more this manuscript:

- Lines 63-65: While cumulative phosphorylation on GPCRs is believed to determine the affinity of β -arrestin interaction with the receptor, any potential explanation for why mutation of a single

phosphorylation site in V₂R (T360A) drastically alters β -arrestin trafficking patterns? Are there additional evidence in the literature of such maximal effect with only a single phosphorylation site mutated for other receptors? It would be interesting to elaborate a bit on this aspect as V₂R^{T360A} is central to the present manuscript.

- In Fig.1B, it would be useful to indicate with an arrow the residue 360 as it takes a few second to figure out where it is.

- The majority of paragraphs from the results start with "In order to". Please use alternative ways to start these paragraphs as it distracts from the interesting story.

- Lines 140-142: The fact that the authors did not observed any measurable effect of Ib30 on G protein-mediated cAMP responses is a bit unexpected. If Ib30 promotes β -arrestin trafficking to early endosomes, we should expect to see more cAMP produced as megaplex formation increases endosomal Gs activation by a GPCR. Can the authors comment on this and maybe briefly clarify this in this section of the results?

- Lines 176-177: Any idea why Ib30 did not have a potentiator effect on ERK1/2 activated by wild-type V₂R if Ib30 increases β -arrestin recruitment to early endosomes?

- Line 244: There is a typo in β arr- β ₂-adaptatin.

- Line 271-273: It would be indeed interesting to explore in future studies whether the effect of Ib30 observed for V₂R^{T360A} is linked to the transition between the partially- and fully-engaged β arr conformations in complex with the receptor. However, before to explore this question, it would be primordial to test if Ib30 rescues other V₂R mutants (mutations on phosphorylated Ser/Thr on the carboxy-terminal), such as the mutants characterised in reference 10.

- Line 581: Usually one star means $p < 0.05$. The authors wrote $*p < 0.01$. Is this a mistake? Same comment for the figure legend corresponding to Fig. S1.

Clarity and context:

The ideas are very easy to follow as they are clearly explained step-by-step and the flow follows a logical sequence. Additionally, sufficient context have been provided to understand the rationale of each experiment.

References:

The references are appropriate.

Reviewer #2 (Remarks to the Author):

Review of "Allosteric modulation of GPCR-induced β -arrestin trafficking and signaling by a synthetic intrabody" by Baidya et al.

In this manuscript the authors present a detailed investigation of the how contact with binding partners controls the conformation and function of β -arrestin 1. First the authors evaluate how the binding of phosphorylated peptides either with or without a key phosphorylated residue (at position T360) or the synthetic intrabody (Ib30) impact arrestin conformation, as measured using a limited trypsin proteolysis assay. These assays suggested subtle conformational differences were induced through arrestin binding as demonstrated by differences in the extent and site of trypsin proteolysis. These differences were evaluated in the context of previously published structural data as well as new molecular dynamics studies. The authors next evaluated functional consequences of the interaction between the synthetic intrabody and beta-arrestin in live cells. Here they found that Ib30 expression facilitates endosomal trafficking upon V2R T360A activation, a very surprising and interesting finding. This effect of Ib30 was shown to correlate with an enhanced interaction between beta-arrestin and beta-adaptin, which is known to facilitate internalization.

Overall the paper is well written, the experiments are carefully performed and interpreted appropriately, and the biological implications of the findings are high. Some of the arguments made regarding structural changes are a bit speculative. The importance of understanding (and controlling) receptor internalization mechanisms is extremely high. As such, I think the manuscript could be suitable for publication provided some major and minor concerns are addressed.

Major comments:

*Receptor internalization (and signaling from internalized compartments) is increasingly being recognized as an important element in determining the biological output of receptor activation. The system described here seems to offer a way to toggle internalization on/off. However, this manuscript doesn't address whether the internalization observed for bArrestin in cells expressing V2R-T360A and Ib30 also correlates with receptor internalization. The authors describe already use a whole cell ELISA assay for looking at receptor levels. This assay (or something related) should be used to see whether

Ib30 also induces receptor internalization for V2R-T360A. This would substantially improve the impact of this manuscript.

*In Figure 3 the authors use luciferase complementation to measure beta-arrestin recruitment to the cell surface. In Figure 4 the authors use BRET to measure arrestin recruitment to the endosome. It would be better if the authors could run the same kind of assay for both plasma membrane and endosomal recruitment. Plasma membrane anchored GFP plasmids are available for BRET.

*In figure 5b the frequency of different angles for interdomain rotation is shown as a result of molecular dynamics simulation for beta arrestin in complex with various partners. Predicting conformations for dynamic proteins is tricky and predicting binding for antibody (fragments) is also tricky. Can the authors provide more evidence that the modeling they performed for bArr1/VwRppT360A/Fab30 is likely to be reflective of reality? As it stands now, I feel the manuscript makes too strong of a conclusion from a modelling experiment that is filled with uncertainty.

Minor comments:

*For all statistical comparisons a one-way ANOVA is used to provide p-values but this test requires post-hoc analysis (like Tukeys) to inform you which groups are actually statistically different from each other. Could the authors specify which post-hoc tests were used? It would also be helpful to specify which two groups are being compared for each p value (I think I can follow but I don't want to have to assume)

*Why is V2RppT360-2- called this? It doesn't have a Thr at position 360?

*For limited proteolysis experiments can the authors demonstrate that the phosphopeptides are not being degraded at different rates?

*In Figure 1c why is there a band labeled Gly(-8) to Arg418 that doesn't appear in Figure 2c?

*What accounts for the reduced ability of ScFv30 to immunoprecipitate beta-arrestin 1/V2RppT360 complex (Figure 2a)? Is the phosphopeptide used at a saturating concentration for beta-arrestin here (what is the KD for this interaction?) Is it because ScFv30 has a lower affinity for beta-arrestin 1/V2RppT30 complex? If this is the case could you explain it explicitly in the text

*In Figure 2e-f what do the dotted lines represent?

*In Figure 3f no error bars are shown although the caption says three independent experiments were performed. Can the authors clarify?

*The meaning of the data presented in the inset in Figure 4b is unclear. Could the authors expand on the explanation of this data or plot it in a different way?

*It seems the level of Ib-CTL/30 is variable in Figure 4c and no conventional loading control is shown. Can the authors comment on this?

*Could the authors comment on whether endogenously expressed beta-arrestins play any role in the observed results for cell-based assays?

*Why is the non-specific band observed in Figure 6A not seen in other blots?

Reviewer #3 (Remarks to the Author):

In their manuscript, the authors investigate the effect of the intrabody30 (Ib30) on the activity state of arrestin bArr1 bound to the V2R. The manuscript builds on previous observations published in Nat. com., that the T360A mutation at the C-terminus of the receptor leads to a loss of arrestin translocation to the endosomal membrane due to a prominent shift in arrestin from active to inactive. In the current manuscript, they investigate the effect of Ib30 binding to bArr1 to reverse this effect. For this purpose, they use various biochemical and biophysical in vitro and in silico methods. In their experimental setup the authors find that Ib30 rescues agonist induced ERK 1/2 MAP kinase activation by the T360A mutation. Moreover, that Ib30 enhances interaction of bArr1 with β 2-adaptin. To the reviewer, the experiments appear well presented and convincing.

In the second part of the manuscript, structural explanations are provided. This portion is of particular importance for the argumentation, as it establishes a reference to the structural level. However, in the opinion of the reviewer, this part is not elaborated far enough and should be deepened or better documented in order to make it accessible and understandable to the broad readership of the journal. The MD simulations indeed seem to have the potential to reproduce and substantiate the observations at a structural level. The interdomain rotation angle used here is a recognized measure for classifying a state as active or inactive. In the present manuscript, the authors use this very measure to support the above hypothesis. The results of three simulation repeats are summarized in Fig. 5B in the form of a histogram. The current presentation of the data appears to the reviewer to be inadequate because significance is not apparent from the provided histograms. However, the data provided to the reviewer on request during the review process allows the following assessment of the analyses presented:

The MD simulations indeed confirm the hypothesis, since within the limitation of non-sufficient sampling, the WT arrestin tends to be in active conformation, the T to A mutation tends to be in inactive conformation, and the latter mutation in complex with intrabody30 (Ib30) again tends to be in active conformation. From the running averages now available to the reviewer, it is clear that the arrestin does not remain in one state in each case, but rather fluctuates between active and inactive, which has not previously been described in this way in the results section. This fluctuation proves that the respective tendency to active or inactive, is not an artifact of a single simulation, but rather and despite the limited simulation time describes an actual property of the system. The reviewer suggests to integrate the histograms into the illustration of the running averages, for example the present illustration turned by 90°, at the end of this. This would already make clear from the figure that the undoubtedly different tendencies are due to an overall limited amount of data - in this case three short simulations - and do not represent the actual ensemble. In doing so, the dimensions of the Y-axes should also be aligned for

all three systems to ensure direct comparability. Furthermore, a space between the individual trajectories should convey even better to the reader that we are dealing with three independent runs and not a continuous simulation. In any case, this limitation, which most simulation work is subject to due to the limiting and costly calculations, should be sufficiently described in the paper.

The reviewer wonders why the structural interpretations from Figs. 2E,F and 5A based on previously published crystal structures, were not supported with MD simulations. On the one hand, the view on static structures does not allow optimal statistics of the contacts, i.e. it is not clear whether the structural interpretations mentioned in the text, which are partly due to side chain movements of single amino acids, are equally observable in a dynamic setup. Second, and more importantly, the described effects of the T to A mutation on the activation or deactivation of arrestin or the effect of the intrabody30 (Ib30) thus do not experience a sufficient structural explanation. A more detailed study of the structures using MD, or further analysis of the existing MD data, should allow the authors to describe the long-range allosteric effects underlying their observations. Naturally, the MD part would thus assume an even more importance in the manuscript.

Finally, the methodology section appears to the reviewer to be in urgent need of revision, as essential information cannot be found here. It was neither helpful for the review, nor does it facilitate the readability of the manuscript, if essential details are not found in the manuscript but in another publication, which is merely referred to. The method used to measure the interdomain rotation in the trajectories is thus not transparent, mainly because the paper they cite does not explicitly state information such as: which axis they use to measure the rotation angle or which reference points (selected residues/atoms or center of mass/center of mass of the selected residues?). Likewise, the information on the PDB-entries, which provided the structure coordinates for the respective simulations, is missing. What changes were made to the structures (mutations, etc.)? Thus, it was laborious for the reviewer to gather this information from other parts of the manuscript. The reader should be spared this effort. A final question arises from the data in the methodology section. The authors used the NVT ensemble for the production runs, which is rather unusual. Why were this ensemble chosen?

In summary, this manuscript describes work with potential interest for the broad readership of the journal. The experimental part of the work seems convincingly elaborated to the reviewer. The structural level is of great importance for the verification of the biochemical analyses and substantial for the understanding of the effects described here. There are still deficiencies here, but these could be solved within the interdisciplinary orientation of the authors. It is important for the reader to understand not only that the T to A mutation has an effect on the dynamic equilibrium of inactive to active arrestin states, but also how this effect comes about structurally and how it can be reversed by binding of intrabody30 (Ib30). At this point, further analysis seems necessary. Finally, the MD part should be described and documented in more detail. The working group around Jana Selent has created a useful basis for sharing MD data with the implementation of GPCRMD. I would expect her to use this

to share the trajectories analyzed here with the reviewers adequately and subsequently make them available to the community in the spirit of the FAIR principles.

Reviewer #4 (Remarks to the Author):

The submitted manuscript by Baidya et al. entitled “Allosteric modulation of GPCR-induced β -arrestin trafficking and signaling by a synthetic intrabody” is a well written manuscript. However, during the entire review process of the manuscript, I had the feeling I have read this story somehow, at least in large part, before.

Major concern:

1) The mutant V2R-T360A and its effect on arrestin recruitment and V2R Trafficking was published before in Ref 10 (by Dwivedi-Agnihotri et al. in 2020 see figure 5 in that paper with arrestin recruitment, trafficking and ERK activation). So what is new here except a bit of ERK stabilization by Ib30

2) What the authors investigate in figure 1 and 2 is the relative conformational change of arrestin by a limited tryptic digest, an established technique since 2006. However, this technique is limited but can show global differences. Since there is no loading control available in these provided gels any conclusion on kinetic effects for the digest will depend on the amount of arrestin added in the individual sample and this is not quantified, at least I did not see this. Hence, this part of discussion is rather weak if not over interpreted.

3) In figure 4 B the V2R wt conditions without and with Ib30 seem to be only shifted in parallel upwards, please take this into consideration in the discussion. The effect might not be as big as described if the two curves would start from the same basal level....

4) Figure 5 is of concern as well. The modeling is nice but the major effect of Ib30 stabilizing the active conformation of arrestin was shown before using Fab30, the parental antibody of Ib30. This was shown by stabilizing NMR signals of arrestin (cited as Ref. 26). Hence, again the novelty is limited since this result was to be expected, although it is great to see this stabilization in living cells and not only NMR.

5) In figure 6 C the V2R T360A conditions without and with Ib30 seem to be only shifted in parallel upwards, this needs to be taken into account. If the curves would start at the same basal level they would be identical and Ib30 had no effect for the T360A mutant...

Minor points:

For this manuscript it would also be helpful to clearly state in the manuscript the differences or similarities between T360A-1 and -2 and when they can be summed up to T360A to follow the authors' throughout the manuscript.

Please make sure that all needed information about statistical analysis is clearly and consistently stated in all figure legends etc. throughout the manuscript.

Line 27, 63/64 vasopressin 2 receptor or vasopressin type 2 receptor

Line 76 previously described intrabody (Ib30)-based sensor (hyphen)

Line 127 additional information to the CONTACT/ACT program would be nice in order to make it more understandable to readers who are not so familiar

Line 211 Please elaborate on the ear-domain of β 2-adaptin for the less conversed readers

Figure 1

- please indicate the starting and end numbers/positions of the shown amino acids in Fig. 1B
- either reduce the shown trypsin digest data in D (because it seems like the data are shown twice for the different trypsin : β arr1 ratios) or make clear why it is important to show both/ point out what is the important difference/message to take from it as a reader
- of note: significance level is set to * $p < 0.1$ instead of 0.05... (which is not the case in Fig.2 for example)

Figure 2

- it would be nice to have the probes in 2C consistently with 1C to make it easier for the reader
- please indicate whether T360-1 or -2 are shown in Fig.2D or whether it is a representation for both, the Fig.2 focusses on T360-1 in E and F, is this also the case in D?
- in figure 2c please also indicate if the big band above 29kDa is the ScFc30 or something else

Figure 3

- since the authors only use dark receptor constructs, they should include at least one control in the supplements where they do not co-express the receptor of interest to clearly show that the observed effect is dependent on the presence of the receptor

Figure 4

- in Figure 4D, the statistical comparison of V2R-T360A + Ib30 to the WT receptor would be also interesting to add statistical relevance to the statement “it robustly enhanced the level of phosphorylated ERK1/2 upon agonist-stimulation, nearly to that of the V2RWT” (line 177/178)

Figure 6

- please include statistical analysis for Fig.6 to support the statement “There was robust interaction between β arr1 and β 2-adaptin for the V2RWT upon agonist-stimulation in the presence of control intrabody, while the response was significantly lower for V2R T360A” (line 221-223) and also in lines 224-227.

Figure S1

- the quantification of the 48kDa band seems to be missing in Fig. S1 and the information of the quantified time point (B)

- of note: significance level is set to * $p < 0.1$ instead of 0.05... (which is not the case in Fig.S2 for example)

Figure S2

- number of independent experiments not indicated

Figure S3

- description of graphs inconsistent, kDa information missing

Figure S4

- was the significance level set to ns $p > 0.1$ or > 0.05 here?

Figure S5

- no statistical analysis of cAMP response and surface expression quantification, but claim in main text “suggesting that the intrabody does not significantly influence agonist-induced $G_{\alpha s}$ coupling (Figure S5A-B).”

Link to supplementary Figure S5B not explicitly clear/mentioned here as the sentence only focusses on cAMP responses

My overall conclusion is that this paper is written well, but the story is rather confirmatory and does not hold any surprises or unforeseeable results.

**Response to reviewers' comments (NCOMMS-22-01576)**

**Reviewer #1:**

**Key results:**

In this study, the authors characterise the ability of intrabody30 (Ib30) to rescue β -arrestin1
translocation to endosomes impaired by a mutation of a phosphorylation site in the carboxy-
terminal region of the vasopressin type 2 receptor (V_2R) previously reported by the same
group. The authors demonstrate that Ib30 also rescues agonist-induced ERK1/2
phosphorylation by the mutant V_2R (V_2R^{T360A}), an event downstream and dependent of β -
arrestin1 translocation to endosomes. Molecular dynamics simulations previously suggested
that V_2R^{T360A} reduces the fraction of active β -arrestin1 conformation and in the present study
elegant in vitro techniques were used to demonstrate that Ib30 rescues β -arrestin1
translocation to endosomes and agonist-induced ERK1/2 phosphorylation by enriching the
active-like conformation population of β -arrestin1. The authors also demonstrate that Ib30
enhances the interaction between β -arrestin1 and β_2 -adaptatin, which is a prominent
mechanism that drives receptor internalisation. They hypothesise that Ib30-mediated increase
of β -arrestin1/ β_2 -adaptatin interaction is the possible mechanistic basis linking active β -
arrestin1 enrichment and improved β -arrestin1 translocation to endosomes induced by Ib30.
Overall these exciting findings provide a proof-of-concept that conformation-specific
intrabodies represent a novel molecular tool to allosterically modulate agonist-induced
trafficking of β -arrestins and therefore constitute an attractive solution to enhance this pathway
in a therapeutic context.

We thank the reviewer immensely for her/his time to review our manuscript so thoroughly and
for her/his positive comments.

**Validity:**

The interpretation and conclusions are based on solid data obtained from both in vitro and
cell-based assays. The authors frequently use several complementary approaches to
measure a given parameter, which strengthens the conclusions and considerably reduces the
risk of conclusions based on artefacts.

We thank the reviewer for her/his positive comments.

**Significance:**

Endosomes represent a fantastic signalling platform where specific cellular functions can only
occur from these intracellular compartments. For some receptors, ERK1/2 activation is an
example, but since the past decade, a growing number of studies report a major functional
role of G protein signalling from endosomes in cellular physiology. As Ib30 induces an
enrichment of fully-active β -arrestins promoting their trafficking to endosomes, this intrabody
could theoretically be used to positively modulate downstream signalling dependent of the
presence of β -arrestins in endosomes by any GPCR of interest (not only V_2R). Supporting this
idea, the authors report that Ib30 efficiently recognises β -arrestin1 in complex with several
native GPCRs (ref. 12, 22). Some studies report that β -arrestin binding to class B receptors in
endosomes potentiates endosomal G protein activation by these receptors. Therefore, by
promoting endosomal trafficking of β -arrestin, Ib30 could
represent an interesting tool to increase endosomal G protein signalling of these receptors,
and consequently selectively potentiate their downstream cellular functions. Additionally, by

its ability to recognise the active conformation of β -arrestin, Ib30 could potentially be utilised
in the context of drug discovery to screen compounds stabilising a receptor conformation
promoting endosomal β -arrestin signalling. Overall, the present study will certainly be of great
significance to the field of G protein-coupled receptor (GPCR) signalling from fundamental
research to investigate the role of endosomal β -arrestin signalling to applied research to
design new therapeutics targeting cellular functions downstream of β -arrestin signalling from
endosomes.

We thank the reviewer for her/his positive comments and appreciation of our work.

**Data and methodology:**

Enough details are provided in the methods for the work to be reproduced. The methodology
is overall elegant and adapted to the questions investigated. The authors combine a well-
balanced ensemble of in vitro techniques, cell-based assays and computational approaches
to answer their questions. In several situations, the authors combine different approaches
together to answer a given question. The work certainly meet the expected state-of-the-art
standards in the field.

We thank the reviewer for her/his positive comments and appreciation of our work.

**Analytic approach:**

The analytical approach is appropriate. However, in Fig. 1D, as data are normalized to
V_2Rpp^{WT} , the % band protection to this condition is always 100. Consequently, the variance
between the 4 experimental groups is not homogenous and therefore an ANOVA cannot be
performed. In that situation, a one-sample t test and comparison to 100 would be the proper
statistical test. Same thing for Fig. 2B, 4D, 6B, S1B, S2B, S3A, and S4A.

We thank the reviewer for her/his positive comments and appreciation of our work.

**Suggested improvements:**

The overall quality of the manuscript is excellent. I do not recommend any additional
experiment. However, I have the following minor concerns to improve even more this
manuscript:

- Lines 63-65: While cumulative phosphorylation on GPCRs is believed to determine the
affinity of β -arrestin interaction with the receptor, any potential explanation for why mutation
of a single phosphorylation site in V_2R (T360A) drastically alters β -arrestin trafficking patterns?
Are there additional evidence in the literature of such maximal effect with only a single
phosphorylation site mutated for other receptors? It would be interesting to elaborate a bit on
this aspect as V_2R^{T360A} is central to the present manuscript.

The dramatic alteration of the \$\beta\$ arr trafficking pattern upon Thr³⁶⁰Ala mutation likely arises from
the disruption of a key salt-bridge with Lys²⁹⁴ in the lariat loop and resulting conformational
change in \$\beta\$ arrs. Following the reviewer's advice, we have now included this information in the
revised text (page 3-4, line 71-73). We have now also cited an additional reference on the
apelin receptor, which also shows a dramatic reduction in \$\beta\$ arr recruitment upon mutation of a
single phosphorylation site (new reference 12 in the revised manuscript).

- In Fig. 1B, it would be useful to indicate with an arrow the residue 360 as it takes a few second
to figure out where it is.

Following reviewer's advice, we have now indicated residue Thr³⁶⁰ in the revised Figure 1B.

- The majority of paragraphs from the results start with "In order to". Please use alternative
ways to start these paragraphs as it distracts from the interesting story.

Following reviewer's advice, we have now revised the text to avoid the repetition of the same
starting sentence in different paragraphs.

- Lines 140-142: The fact that the authors did not observed any measurable effect of Ib30 on
G protein-mediated cAMP responses is a bit unexpected. If Ib30 promotes β -arrestin
trafficking to early endosomes, we should expect to see more cAMP produced as megaplex
formation increases endosomal Gs activation by a GPCR. Can the authors comment on this
and maybe briefly clarify this in this section of the results?

This is an interesting point. We are using the GloSensor assay to measure cAMP responses,
and this assay saturates rather quickly due to massive amplification of the signal. This may
be a plausible reason for why we could not detect a difference in the cAMP response.
However, there may be additional possibilities for this interesting observation that should be
explored further in future studies. Following reviewer's advice, we have now discussed this in
the revised manuscript (page 14, line 326-331).

- Lines 176-177: Any idea why Ib30 did not have a potentiator effect on ERK1/2 activated by
wild-type V₂R if Ib30 increases β -arrestin recruitment to early endosomes?

This is also an interesting point. In case of V₂R^{WT}, the starting point of the curve is also up-
shifted and therefore, the net potentiation of the endosomal trafficking is lesser than V₂R^{T360A}.
We have now clarified this point further by plotting the data differently i.e. differences in BRET
signal between the lowest and highest agonist dose (Figure 6C and 6F). Moreover, as we are
measuring ERK1/2 phosphorylation at the 5min time-point, where the response is typically
maximal, minor potentiation of ERK1/2 may not be apparent. Following reviewer's advice, we
have now discussed this in the revised manuscript (page 13, line 316-322).

- Line 244: There is a typo in β arr- β ₂-adaptatin.

This has been corrected.

- Line 271-273: It would be indeed interesting to explore in future studies whether the effect of
Ib30 observed for V₂R^{T360A} is linked to the transition between the partially- and fully-engaged
β arr conformations in complex with the receptor. However, before to explore this question, it
would be primordial to test if Ib30 rescues other V₂R mutants (mutations on phosphorylated
Ser/Thr on the carboxy-terminal), such as the mutants characterised in reference 10.

This is an interesting point too. In fact, the other V₂R mutants characterized in reference 10
display either a wild-type-like pattern of β arr trafficking (e.g. V₂R^{S357A} and V₂R^{T359A}) or, a
complete loss of β arr trafficking (e.g. V₂R^{S362A/S363A/S364A}). Therefore, we focused our attention
on V₂R^{T360A} in the current manuscript, which displays robust β arr binding but a dramatically
altered trafficking profile compared to V₂R^{WT}. Following reviewer's advice, we have now
mentioned this in the revised manuscript (page 4, line 73-76).

- Line 581: Usually one star means $p < 0.05$. The authors wrote $*p < 0.01$. Is this a mistake?
Same comment for the figure legend corresponding to Fig. S1.

We thank the reviewer for pointing this out. We have corrected it in the revised manuscript.

**Clarity and context:**

The ideas are very easy to follow as they are clearly explained step-by-step and the flow
follows a logical sequence. Additionally, sufficient context have been provided to understand
the rationale of each experiment.

We thank the reviewer for her/his positive comments.

**References:**

The references are appropriate.

We thank the reviewer for her/his positive comments.

**Reviewer #2:**

Review of “Allosteric modulation of GPCR-induced β -arrestin trafficking and signaling by a
synthetic intrabody” by Baidya et al.

In this manuscript the authors present a detailed investigation of the how contact with binding
partners controls the conformation and function of β -arrestin 1. First the authors evaluate how
the binding of phosphorylated peptides either with or without a key phosphorylated residue (at
position T360) or the synthetic intrabody (Ib30) impact arrestin conformation, as measured
using a limited trypsin proteolysis assay. These assays suggested subtle conformational
differences were induced through arrestin binding as demonstrated by differences in the extent
and site of trypsin proteolysis. These differences were evaluated in the context of previously
published structural data as well as new molecular dynamics studies. The authors next
evaluated functional consequences of the interaction between the synthetic intrabody and
beta-arrestin in live cells. Here they found that Ib30 expression facilitates endosomal trafficking
upon V_2R^{T360A} activation, a very surprising and interesting finding. This effect of Ib30 was
shown to correlate with an enhanced interaction between beta-arrestin and beta-adaptin,
which is known to facilitate internalization.

We thank the reviewer immensely for her/his time to review our manuscript so thoroughly and
for her/his positive comments.

Overall, the paper is well written, the experiments are carefully performed and interpreted
appropriately, and the biological implications of the findings are high. Some of the arguments
made regarding structural changes are a bit speculative. The importance of understanding
(and controlling) receptor internalization mechanisms is extremely high. As such, I think the
manuscript could be suitable for publication provided some major and minor concerns are
addressed.

We thank the reviewer for her/his positive comments and important feedback.

Major comments:

*Receptor internalization (and signaling from internalized compartments) is increasingly being
recognized as an important element in determining the biological output of receptor activation.
The system described here seems to offer a way to toggle internalization on/off. However, this
manuscript doesn't address whether the internalization observed for bArrestin in cells
expressing $V_2R-T360A$ and Ib30 also correlates with receptor internalization. The authors
describe already use a whole cell ELISA assay for looking at receptor levels. This assay (or
something related) should be used to see whether Ib30 also induces receptor internalization
for $V_2R-T360A$. This would substantially improve the impact of this manuscript.

This is a very interesting point. Following the reviewer's suggestion, we measured the
endocytosis and trafficking of V_2R^{T360A} using whole cell ELISA and NanoBiT assays. While we
expected that receptor endocytosis would follow a similar pattern as β arr1 trafficking, we
surprisingly observed that V_2R^{T360A} internalizes and localizes to endosomes as efficiently as
V_2R^{WT} (please see Figure R2.1 below). These data are quite intriguing as they seem to
suggest separate trafficking of V_2R and β arr1 after agonist-stimulation. We believe however
that this interesting lead requires substantial additional investigation as a follow up study, and
we now have discussed this point in the revised manuscript (page 14, line 330-334).

*In Figure 3 the authors use luciferase complementation to measure beta-arrestin recruitment
 to the cell surface. In Figure 4 the authors use BRET to measure arrestin recruitment to the
 endosome. It would be better if the authors could run the same kind of assay for both plasma
 membrane and endosomal recruitment. Plasma membrane anchored GFP plasmids are
 available for BRET.

Following the reviewer's suggestion, we have now measured surface and total recruitment of
 β arr1 as well as the endosomal trafficking of β arr1 using NanoBiT assays. We have now
 included these data as Figure 4A-D (presented below as Figure R2.2) and Figure 6 and S5
 (presented below as Figure R2.2). The endosomal trafficking data using NanoBiT assay
 essentially recapitulate the pattern observed using the BRET assay.

*In figure 5b the frequency of different angles for interdomain rotation is shown as a result of
 molecular dynamics simulation for beta arrestin in complex with various partners. Predicting
 conformations for dynamic proteins is tricky and predicting binding for antibody (fragments) is
 also tricky. Can the authors provide more evidence that the modeling they performed for
 bArr1/VwRppT360A/Fab30 is likely to be reflective of reality? As it stands now, I feel the
 manuscript makes too strong of a conclusion from a modelling experiment that is filled with
 uncertainty.

We completely understand the point made by the reviewer. However, we would like to
 underscore that our simulation studies are based on previously determined high-resolution
 crystal structures of βarr1- $V_2R^{WT}/V_2R^{T360-1}/Fab30$ complexes (PDB: 4JQI and 7DFA). It is
 important to note that these crystal structures have provided an excellent understanding of the
 Fab30 binding interface on βarr1 and the inter-domain rotation, which guided our molecular
 dynamics simulation studies. In fact, our previous study has revealed that $T^{360}A$ mutation in
 V_2R shifted the conformational equilibrium of βarr1 towards an inactive state characterized by

smaller inter-domain rotation, an observation we have recapitulated in the current study as
well. Moreover, another recent study based on V₂Rpp-βarr1-Fab30 structure has reported that
the removal of the phosphopeptide resulted in βarr1 inactivation with respect to inter-domain
rotation angle. As the binding interface of Fab30 spans both, the N-domain and C-domain in
βarr1, its positive effect on inter-domain rotation, as observed in our simulation studies, is
highly anticipated. Finally, a recent study has measured the effect of Fab30 on βarr1
conformation using solution NMR methodology, and converged onto a similar conclusion in
terms of stabilizing effect of Fab30 on βarr1 activation as observed in our simulation study
(30). Therefore, we believe that our simulation data provide important insights into the
allosteric effect of Fab30 on βarr1 activation with respect to inter-domain rotation that can be
linked to βarr1 trafficking and signaling. Still however, following reviewer's feedback, we have
now included a brief discussion on this in the revised manuscript (page 11-12, line 271-276).

Minor comments:

*For all statistical comparisons a one-way ANOVA is used to provide p-values but this test
requires post-hoc analysis (like Tukeys) to inform you which groups are actually statistically
different from each other. Could the authors specify which post-hoc tests were used? It would
also be helpful to specify which two groups are being compared for each p value (I think I can
follow but I don't want to have to assume).

We thank the reviewer for pointing this out, and we have now included this information in the
figures and figure legends in the revised manuscript.

*Why is V₂RppT360-2⁻ called this? It doesn't have a Thr at position 360?

V₂Rpp^{T360-2} has an Alanine at position 360 while V₂Rpp^{T360-1} has a non-phosphorylated
Threonine. In V₂Rpp^{WT}, there is a phosphorylated Threonine at position 360. We have
mentioned this explicitly in the revised manuscript (page 4, line 90-94) and also indicated the
position 360 with an arrow in Figure 1B.

*For limited proteolysis experiments can the authors demonstrate that the phosphopeptides
are not being degraded at different rates?

This is an interesting point. As the phosphopeptides are only 29 amino acid long, it is difficult
to follow their proteolysis patterns using SDS-PAGE, and it will require more sophisticated
approaches(e.g. mass spectrometry). However, we have analyzed the phosphopeptide
sequence *in-silico*, and as presented in Figure R2.3 below, the trypsin cleavage sites are
present only at the edges in these peptides. In other words, even if the peptides were being
degraded differentially by trypsin, it will not influence the proteolysis pattern of βarr1 as the
core segment of the phosphopeptides harbouring the phosphorylated residues would remain
intact.

↓ ↓ ↓

ARGRIPPSLG PQDESCTTASSSLAKDTSS

**Figure R2.3. Trypsin cleavage sites in V₂Rpp.** The amino acid sequence of V₂Rpp^{WT} was analyzed using PeptideCutter (Expasy) with trypsin as the enzyme, and the potential cleavage sites are indicated with red arrows. The phosphorylated Ser/Thr residues in the peptide are indicated in blue.

*In Figure 1c why is there a band labeled Gly(-8) to Arg418 that doesn't appear in Figure 2c?

The 48kDa band is not apparent in Figure 2C as the time-point (post-trypsin addition) for this
experiment is 30min (in contrast with Figure 1C where the time-point is 5min), at which the
48kDa band is completely digested. We have now indicated the time points in the respective
figure legends to clarify this point.

*What accounts for the reduced ability of ScFv30 to immunoprecipitate beta-arrestin
1/V2RppT360 complex (Figure 2a)? Is the phosphopeptide used at a saturating concentration
for beta-arrestin here (what is the KD for this interaction?) Is it because ScFv30 has a lower
affinity for beta-arrestin 1/V2RppT360 complex? If this is the case could you explain it explicitly
in the text?

This is an interesting point and we have observed this pattern for both, ScFv30 and Fab30. In
these co-IP experiments, we used phosphopeptides at two different molar excess ratios i.e.
1:10 and 1:50 as indicated in the figures, and therefore, phosphopeptides are indeed at
saturating concentrations. Although we have not measured the Kd value for the interaction of
the phosphopeptides with β arr1, it is plausible that Fab30/ScFv30 have lower affinities for
β arr1-V₂Rpp^{T360-1/2} complexes compared to β arr1-V₂Rpp^{WT} complex. Following reviewer's
suggestion, we have now mentioned this in the revised manuscript (page 5, line 119-121).

*In Figure 2e-f what do the dotted lines represent?

The dotted lines represent hydrogen bonds and polar interactions between the indicated
residues, and we have now mentioned this in the revised figure legend.

*In Figure 3f no error bars are shown although the caption says three independent
experiments were performed. Can the authors clarify?

Here, we have monitored the localization of β arr1 using confocal microscopy in different fields
of view and manually scored β arr1 localization from more than 500 cells for each condition.
These data are presented as % normalized i.e. what % of total cells display membrane
localization of β arr1 vs. punctate localization. Here, three independent replicates represent
three different transfections on different days, their confocal imaging and manual scoring of
β arr1 localization from pool of cells from these three experiments. Following reviewer's
suggestion, we have now clarified this in the revised manuscript (page 9, line 202-203 and
page 21-22, line 519-527 and revised figure legend of Figure 5F).

*The meaning of the data presented in the inset in Figure 4b is unclear. Could the authors
expand on the explanation of this data or plot it in a different way?

We regret the lack of clarity, and we have now plotted this data differently. We have essentially
shown the change in BRET signal (Δ BRET) for all four conditions in Figure 6C. The key point
in this graph is to demonstrate the effect of Ib30 on endosomal trafficking of β arr1 in terms of
maximal agonist-induced response i.e. change in BRET signal. We have also clarified this
point in the revised figure legend.

*It seems the level of Ib-CTL/30 is variable in Figure 4c and no conventional loading control is
shown. Can the authors comment on this?

Although we have expression normalized Ib-CTL and Ib30, there is a slight variation in their
relative expression levels in some experiments. But, despite a slightly lower expression level,

Ib30 exhibits a robust positive effect on ERK1/2 activation, which further strengthens the point.
 We had not included conventional loading control e.g. β -actin as the total-ERK1/2 blot itself
 serves as a loading controls. Still however, following reviewer's suggestion, we have now
 repeated the experiment and show β -actin loading control as well (presented below as Figure
 R2.4; and included as Figure 7A in the revised manuscript).

*Could the authors comment on whether endogenously expressed beta-arrestins play any role
 in the observed results for cell-based assays?

This is an interesting point. However, in most of the cellular experiments except ERK1/2 assay,
 we are using exogenous β arr1 that is engineered for the corresponding assay (e.g.
 SmBiT/LgBiT-tagged in NanoBiT assay, mCherry/mYFP-tagged in confocal imaging and R-
 Luc-tagged in BRET assay). Therefore, the measured responses arise specifically from the
 exogenous β arrs and not from endogenous β arrs. However, we cannot rule out the possibility
 that endogenous β arrs may compete for the receptor upon agonist-stimulation but that is likely
 to be similar across different conditions, if any.

*Why is the non-specific band observed in Figure 6A not seen in other blots?

We thank the reviewer for pointing this out. In this experiment, we have used lysate from S9
 cells expressing V₂R^{T360A} while in other experiments (Figure 7A and Figure 9E), we have used
 HEK-293 cells. We have observed this non-specific band in anti-HA blots when using lysate
 from S9 cells in other experiments as well in our laboratory. We have now clarified in the
 revised figure legend.

**Reviewer #3:**

In their manuscript, the authors investigate the effect of the intrabody30 (Ib30) on the activity
state of arrestin bArr1 bound to the V2R. The manuscript builds on previous observations
published in Nat. com., that the T360A mutation at the C-terminus of the receptor leads to a
loss of arrestin translocation to the endosomal membrane due to a prominent shift in arrestin
from active to inactive. In the current manuscript, they investigate the effect of Ib30 binding to
bArr1 to reverse this effect. For this purpose, they use various biochemical and biophysical in
vitro and in silico methods. In their experimental setup the authors find that Ib30 rescues
agonist induced ERK 1/2 MAP kinase activation by the T360A mutation. Moreover, that Ib30
enhances interaction of bArr1 with b2-adaptin. To the reviewer, the experiments appear well
presented and convincing.

We thank the reviewer for her/his positive comments.

In the second part of the manuscript, structural explanations are provided. This portion is of
particular importance for the argumentation, as it establishes a reference to the structural level.
However, in the opinion of the reviewer, this part is not elaborated far enough and should be
deepened or better documented in order to make it accessible and understandable to the
broad readership of the journal. The MD simulations indeed seem to have the potential to
reproduce and substantiate the observations at a structural level. The interdomain rotation
angle used here is a recognized measure for classifying a state as active or inactive. In the
present manuscript, the authors use this very measure to support the above hypothesis. The
results of three simulation repeats are summarized in Fig. 5B in the form of a histogram. The
current presentation of the data appears to the reviewer to be inadequate because significance
is not apparent from the provided histograms. However, the data provided to the reviewer on
request during the review process allows the following assessment of the analyses presented:
The MD simulations indeed confirm the hypothesis, since within the limitation of non-sufficient
sampling, the WT arrestin tends to be in active conformation, the T to A mutation tends to be
in inactive conformation, and the latter mutation in complex with intrabody30 (Ib30) again
tends to be in active conformation. From the running averages now available to the reviewer,
it is clear that the arrestin does not remain in one state in each case, but rather fluctuates
between active and inactive, which has not previously been described in this way in the results
section. This fluctuation proves that the respective tendency to active or inactive, is not an
artifact of a single simulation, but rather and despite the limited simulation time describes an
actual property of the system. The reviewer suggests to integrate the histograms into the
illustration of the running averages, for example the present illustration turned by 90°, at the
end of this. This would already make clear from the figure that the undoubtedly different
tendencies are due to an overall limited amount of data - in this case three short simulations -
and do not represent the actual ensemble. In doing so, the dimensions of the Y-axes should
also be aligned for all three systems to ensure direct comparability. Furthermore, a space
between the individual trajectories should convey even better to the reader that we are dealing
with three independent runs and not a continuous simulation. In any case, this limitation, which
most simulation work is subject to due to the limiting and costly calculations, should be
sufficiently described in the paper.

We very much appreciate reviewer's time and effort to carefully review our data, and provide
constructive feedback to strengthen our manuscript. Following reviewer's suggestion, we have
now integrated the histograms of the inter-domain rotation angle into the illustration of their
running averages (please see Figure R3.1 below). This figure is now included as Figure 8B-D
in the revised manuscript.

Regarding sampling limitations of MD simulation, we agree with the reviewer that current
simulation times may have limitations in obtaining a converged sampling of the conformational
space of β arrs. To clarify this point to the readers, we have now included a brief discussion of
this the revised manuscript (page 11-12, line 271-276). Moreover, to further support our
findings, we have now increased the sampling to an accumulated time of 10 μ s per system
(i.e. 5 runs of 2 μ s each), which strengthens our observations. We have revised the main text
(line 271-272), method section (page 25, line 597-600) and the corresponding figure legend
(Figure 8B) accordingly.

The reviewer wonders why the structural interpretations from Figs. 2E, F and 5A based on
previously published crystal structures, were not supported with MD simulations. On the one
hand, the view on static structures does not allow optimal statistics of the contacts, i.e. it is not
clear whether the structural interpretations mentioned in the text, which are partly due to side
chain movements of single amino acids, are equally observable in a dynamic setup. Second,
and more importantly, the described effects of the T to A mutation on the activation or
deactivation of arrestin or the effect of the intrabody30 (Ib30) thus do not experience a
sufficient structural explanation. A more detailed study of the structures using MD, or further
analysis of the existing MD data, should allow the authors to describe the long-range allosteric
effects underlying their observations. Naturally, the MD part would thus assume an even more
importance in the manuscript.

The reviewer makes an interesting point here. The structural snapshots included in Figure 3A
(previously 2E-F) were primarily presented to indicate the orientation/positioning of Arg²⁸⁵ and
Arg¹⁸⁸, two trypsin cleavage sites in β arr1 resulting in 32kDa and 21kDa band, respectively,
based on the previously published crystal structures. Similarly, the structural snapshot in
Figure 8A (previously 5A) were presented to highlight the repositioning of the proximal
segment of V₂Rpp and the interaction of T³⁵⁹/T³⁶⁰ in the two crystal structures i.e. V₂Rpp^{WT}-
bound and V₂Rpp^{T360-1}-bound β arr1.

Following the reviewer's suggestion, we have now carried out additional simulations to
sample the conformation of Arg²⁸⁵ and Arg¹⁸⁸ under three different conditions i.e. basal β arr1,
V₂Rpp^{WT}-bound β arr1 and V₂Rpp^{T360-1}-bound β arr1 (please see Figure R3.2 below). These
simulations essentially suggest that the overall conformational space sampled by these two
residues in V₂Rpp^{WT}-bound β arr1 and V₂Rpp^{T360-1}-bound β arr1 states are mostly similar. This
observation matches well with our experimental observations (Figure 2E), where the WT and
mutant peptides induce identical patterns of proteolysis with respect to the 32kDa and 21kDa
bands. We thank the reviewer for making this excellent suggestion that has helped us
rationalize the experimental data better. We have now included these simulation data as
Figure 3B in the revised manuscript.

In addition, we have also expanded the corresponding results sections to include more
mechanistic insight by linking the previous observations with those uncovered in the current
study (page 6-7, line 144-151).

Finally, the methodology section appears to the reviewer to be in urgent need of revision, as
essential information cannot be found here. It was neither helpful for the review, nor does it
facilitate the readability of the manuscript, if essential details are not found in the manuscript
but in another publication, which is merely referred to. The method used to measure the
interdomain rotation in the trajectories is thus not transparent, mainly because the paper they
cite does not explicitly state information such as: which axis they use to measure the rotation
angle or which reference points (selected residues/atoms or center of mass/center of mass of
the selected residues?).

We thank the reviewer for pointing this out, and we have now included the corresponding information in the method section of the revised manuscript (page 25, line 602-609).

Likewise, the information on the PDB-entries, which provided the structure coordinates for the respective simulations, is missing. What changes were made to the structures (mutations, etc.)? Thus, it was laborious for the reviewer to gather this information from other parts of the manuscript. The reader should be spared this effort.

We thank the reviewer for pointing this out, and we have now included the corresponding information in the method section of the revised manuscript (page 25, line 604-606).

A final question arises from the data in the methodology section. The authors used the NVT ensemble for the production runs, which is rather unusual. Why were this ensemble chosen?

The choice of the NVT ensemble was primarily motivated by the simulation package we used ACEMD [PMID: 26609855]). As ACEMD only enables the usage of the Berendsen barostat, it is a good choice for equilibration but not production runs, ACEMD users switch to an NVT ensemble for production runs. Although an NPT ensemble is intuitively more correct for production runs of biological systems, ACEMD developers state that “*With the system sizes which are achievable nowadays, it is not necessary to have a pressure control in the production runs (for large number of atoms all ensembles are equivalent, statistically)*”.

In summary, this manuscript describes work with potential interest for the broad readership of the journal. The experimental part of the work seems convincingly elaborated to the reviewer. The structural level is of great importance for the verification of the biochemical analyses and substantial for the understanding of the effects described here. There are still deficiencies here, but these could be solved within the interdisciplinary orientation of the authors. It is important for the reader to understand not only that the T to A mutation has an effect on the dynamic equilibrium of inactive to active arrestin states, but also how this effect comes about structurally and how it can be reversed by binding of intrabody30 (Ib30). At this point, further analysis seems necessary. Finally, the MD part should be described and documented in more detail.

We thank the reviewer for his/her positive comments and constructive suggestions. As mentioned above against specific comments/suggestions, we have now addressed all the points raised by the reviewer in the revised manuscript.

The working group around Jana Selent has created a useful basis for sharing MD data with the implementation of GPCRMD. I would expect her to use this to share the trajectories analyzed here with the reviewers adequately and subsequently make them available to the community in the spirit of the FAIR principles.

We most certainly support data sharing and transparency, and following reviewer’s excellent suggestion, we have submitted our simulation data to the GPCRMD portal (<https://submission.gpcrmd.org/dynadb/publications/1486/>) and mentioned this in the revised manuscript (page 25, line 608-609).

**Reviewer #4 (Remarks to the Author):**

The submitted manuscript by Baidya et al. entitled “Allosteric modulation of GPCR-induced β -
arrestin trafficking and signaling by a synthetic intrabody” is a well written manuscript.
However, during the entire review process of the manuscript, I had the feeling I have read this
story somehow, at least in large part, before.

Major concern:

1) The mutant V2R-T360A and its effect on arrestin recruitment and V2R Trafficking was
published before in Ref 10 (by Dwivedi-Agnihotri et al. in 2020 see figure 5 in that paper with
arrestin recruitment, trafficking and ERK activation). So what is new here except a bit of ERK
stabilization by Ib30?

We thank the reviewer for her/his careful and critical reading of our manuscript. While in the
earlier study, we characterized the decisive contribution of a single phosphorylation site in the
receptor on β arr1 recruitment, trafficking and ERK1/2 activation, the current study is focused
on how an intrabody is able to allosterically modulate β arr1 trafficking and rescue ERK1/2
activation. Moreover, we provide a mechanistic basis for this allosteric effect exerted by the
intrabody by demonstrating, for the first time, that it can stabilize β arr1- β 2-adaptin interaction.
Furthermore, simulation-based structural analysis also underscores, for the first time, how
fine-tuning inter-domain rotation may influence β arr1 trafficking and ERK1/2 activation, an
insight with broad implications for better understanding the GPCR- β arr interactions and
signaling paradigms. Finally, our study presents a proof-of-principle paradigm to demonstrate
that intrabodies can be utilized to positively modulate functional responses mediated by β arrs
in the context of GPCR activation.

2) What the authors investigate in figure 1 and 2 is the relative conformational change of
arrestin by a limited tryptic digest, an established technique since 2006. However, this
technique is limited but can show global differences. Since there is no loading control available
in these provided gels any conclusion on kinetic effects for the digest will depend on the
amount of arrestin added in the individual sample and this is not quantified, at least I did not
see this. Hence, this part of discussion is rather weak if not over interpreted.

We understand reviewer’s point but there appears to be a slight misunderstanding here. In
these experiments, we prepare β arr1+V₂Rpp samples in parallel, and took out an aliquot
before adding trypsin. Subsequently, we add trypsin to the samples followed by incubation at
37°C for 5min, and take out another volume-normalized aliquot (i.e. accounting for slight
volume change with trypsin addition). We did the same for all samples (i.e. apo-condition and
phosphopeptide-bound) in parallel, and then run trypsin-digested samples for all conditions
side-by-side. Still however, we have now repeated the experiment under apo and V₂Rpp^{WT}
conditions, and ran samples before and after proteolysis for each condition side-by-side to
demonstrate equal starting material in each condition (Figure R4.1). As evident from the band
intensity, the starting amount of protein is comparable for each time-point.

3) In figure 4 B the V2R wt conditions without and with Ib30 seem to be only shifted in parallel
 upwards, please take this into consideration in the discussion. The effect might not be as big
 as described if the two curves would start from the same basal level....

We thank the reviewer for pointing this out, which allows us to further clarify this in the main
 text. In fact, we did not focus here on the V_2R^{WT} , which as the reviewer correctly points out,
 does not change significantly between Ib-CTL and Ib30 conditions, if the parallel shift is taken
 into account. Instead, we underscore the effect of Ib30 on V_2R^{T360A} mutant; where the basal
 between Ib-CTL and Ib30 conditions are identical but the maximal responses are dramatically
 different (compare the line with blue circle to blue square symbols). We have now elaborated
 on this further in the main text (page 9, line 212-215) and also plotted the Δ BRET data
 differently to make this point better (Figure 6C in the revised manuscript).

4) Figure 5 is of concern as well. The modeling is nice but the major effect of Ib30 stabilizing
the active conformation of arrestin was shown before using Fab30, the parental antibody of
Ib30. This was shown by stabilizing NMR signals of arrestin (cited as Ref. 26). Hence, again
the novelty is limited since this result was to be expected, although it is great to see this
stabilization in living cells and not only NMR.

We thank the reviewer for appreciating our modelling and cellular data. Regarding the aspect
of novelty, we would like to underscore that in the previous study, we had reported that the
inter-domain rotation induced in β arr1 upon V_2R^{T360A} mutation is significantly reduced
compared to that by the V_2R^{WT} . We had not explored, or commented on, the influence of Fab30
(or, ScFv30) on the inter-domain-rotation, an aspect that we probe and describe in the current
manuscript for the first time. The NMR data that the reviewer appropriately refers to,
demonstrates the positive effect of Fab30 on promoting a fully-engaged-like conformation of
β arr1 upon interaction with a chimeric β_2 -adrenergic receptor (referred to as β_2V_2R). On the
other hand, the modelling data included in the current study uncovered a positive allosteric
effect of Ib30 on inter-domain rotation in β arr1 for a single phospho-site mutant of the V_2R .

5) In figure 6 C the V2R T360A conditions without and with Ib30 seem to be only shifted in
parallel upwards, this needs to be taken into account. If the curves would start at the same
basal level they would be identical and Ib30 had no effect for the T360A mutant...

We agree with the reviewer but there appears to be another misunderstanding between us. In
fact, our co-IP data (presented in Figure 9A) suggest that ScFv30 enhances the basal
interaction between β arr1 and β_2 -adaptin. This is also reflected in cellular data using BRET
(Figure 9C) where we observe an enhanced basal signal for both, V_2R^{WT} and V_2R^{T360A} in
presence of Ib30 (compared to Ib-CTL). Based on this, we designed the Ib30 titration
experiment presented in Figure 9E, where we observed a saturable response in β arr1- β_2 -
adaptin interaction with increasing Ib30 expression level. Taken together, these data establish
the positive effect of ScFv30/Ib30 on β arr1- β_2 -adaptin interaction, and we have now clarified
this in the revised manuscript (page 12-13, line 296-300).

Minor points:

For this manuscript it would also be helpful to clearly state in the manuscript the differences
or similarities between T360A-1 and -2 and when they can be summed up to T360A to follow
the authors' throughout the manuscript.

We thank the reviewer for mentioning this. We refer to the phospho-peptides in Figure 1 and
2 as V_2Rpp^{T360-1} and V_2Rpp^{T360-2} , and we refer to the receptor mutant as V_2R^{T360A} . We have
now clarified this in the revised manuscript (page 4, line 90-94).

Please make sure that all needed information about statistical analysis is clearly and
consistently stated in all figure legends etc. throughout the manuscript.

We have carefully revised the figure legends to include all the information related to statistical
analysis.

Line 27, 63/64 vasopressin 2 receptor or vasopressin type 2 receptor

We have corrected this in the revised manuscript.

Line 76 previously described intrabody (Ib30)-based sensor (hyphen).

We have corrected this in the revised manuscript.

Line 127 additional information to the CONTACT/ACT program would be nice in order to make
it more understandable to readers who are not so familiar.

We have now included additional details about this in the revised manuscript (page 6, line
140-143).

Line 211 Please elaborate on the ear-domain of β 2-adaptin for the less conversed readers.

We have corrected this in the revised manuscript (page 12, line 281-285).

Figure 1

- please indicate the starting and end numbers/positions of the shown amino acids in Fig. 1B

We have now included this information in revised Figure 1B.

- either reduce the shown trypsin digest data in D (because it seems like the data are shown
twice for the different trypsin: β arr1 ratios) or make clear why it is important to show both/ point
out what is the important difference/message to take from it as a reader.

We have included the limited proteolysis data at two different trypsin: β arr1 ratio in Figure 1D
in order to better highlight the differences between V_2Rpp and $V_2Rpp^{T360-1/2}$. For example, the
difference in the intensity of 48kDa and 47kDa bands are visualized better at 1:50 ratio while
the difference in the 32kDa and 21kDa bands are visualized better at 1:25 ratio. We have now
explained this in the revised manuscript (page 5, line 102-104).

- of note: significance level is set to * $p < 0.1$ instead of 0.05... (which is not the case in Fig.2
for example)

We have now corrected this in the revised manuscript.

Figure 2

- it would be nice to have the probes in 2C consistently with 1C to make it easier for the reader

We have now corrected this in the revised manuscript.

- please indicate whether T360-1 or -2 are shown in Fig.2D or whether it is a representation
for both, the Fig.2 focusses on T360-1 in E and F, is this also the case in D?

Figure 2E reflects the pattern common for both, V_2Rpp^{T360-1} and V_2Rpp^{T360-2} . We have now
corrected this in the revised figure and the figure legend.

- in figure 2c please also indicate if the big band above 29kDa is the ScFc30 or something
else

This band is indeed ScFv30, and we have now indicated it in the revised Figure 2E.

Figure 3

- since the authors only use dark receptor constructs, they should include at least one control
 in the supplements where they do not co-express the receptor of interest to clearly show that
 the observed effect is dependent on the presence of the receptor.

In these experiments, we are measuring the response (change in luminescence signal in
 Figure 5B and localization of β arr1 in Figure 5C and E) upon stimulation of cells with agonist.
 Therefore, the observed effects should clearly be based on receptor activation. Still however,
 following reviewer's suggestion, we have now repeated these experiments without receptor
 expression. As presented below in Figure R4.2, we do not observe the corresponding
 responses in cells lacking the receptor.

Figure 4

- in Figure 4D, the statistical comparison of V2R-T360A + Ib30 to the WT receptor would be
also interesting to add statistical relevance to the statement “it robustly enhanced the level of
phosphorylated ERK1/2 upon agonist-stimulation, nearly to that of the V2RWT” (line 177/178)

Following reviewer’s suggestion, we have now compared Ib30 conditions for V₂R^{WT} and
V₂R^{T360A} and included this in the revised Figure 7D.

Figure 6

- please include statistical analysis for Fig.6 to support the statement “There was robust
interaction between βarr1 and β2-adaptin for the V2RWT upon agonist-stimulation in the
presence of control intrabody, while the response was significantly lower for V2R T360A” (line
221-223) and also in lines 224-227.

Following reviewer’s suggestion, we have now compared Ib-CTL conditions for V₂R^{WT} and
V₂R^{T360A} and included this in the revised Figure 9B. We have also compared the basal and
maximal response for βarr1-β2-adaptin interaction in the BRET assay (Figure 9C) and present
this analysis in Figure 9D.

Figure S1

- the quantification of the 48kDa band seems to be missing in Fig. S1 and the information of
the quantified time point (B)

We have now included the quantification of the 48kDa band in the revised Figure S1 and
indicated the time-point in the figure legend.

- of note: significance level is set to * p < 0.1 instead of 0.05... (which is not the case in Fig.S2
for example)

We have now corrected this in the revised figure legend.

Figure S2

- number of independent experiments not indicated.

We have now corrected this in the revised figure legend.

Figure S3

- description of graphs inconsistent, kDa information missing

We have now corrected this in the revised figure legend.

Figure S4

- was the significance level set to ns p> 0.1 or > 0.05 here?

The significance level for ns was set to p>0.05, and we have now included this in the revised
figure legend.

Figure S5

- no statistical analysis of cAMP response and surface expression quantification, but claim in
main text “suggesting that the intrabody does not significantly influence agonist-induced Gas
coupling (Figure S5A-B).”

Following reviewer’s suggestion, we have now carried out statistical analysis on these data
and included this information in the revised figure and figure legend.

Link to supplementary Figure S5B not explicitly clear/mentioned here as the sentence only
focusses on cAMP responses

Figure S6B shows the data for surface expression of the receptor constructs in the GloSensor
assay (Figure S6A). We have now clarified this in the revised manuscript (page 8, line 184-
187).

REVIEWERS' COMMENTS

Reviewer #1 (Remarks to the Author):

Almost all the recommended improvements have been done in the new version of the manuscript, except one. It seems that the authors forgot to address the concern about the analytic approach (see lines 62 to 67 in the rebuttal). Consequently I recommend this manuscript to be published conditionally to addressing this concern.

Reviewer #2 (Remarks to the Author):

The authors have done a thorough job of responding to my comments. I think the revised manuscript is well-suited for publication. There is one author response on which I request clarification.

Author response to reviewer:

This is a very interesting point. Following the reviewer's suggestion, we measured the
endocytosis and trafficking of V2RT360A using whole cell ELISA and NanoBIT assays. While
we expected that receptor endocytosis would follow a similar pattern as α arr1 trafficking, we
surprisingly observed that V2RT360A internalizes and localizes to endosomes as efficiently as
V2RWT (please see Figure R2.1 below). These data are quite intriguing as they seem to
suggest separate trafficking of V2R and α arr1 after agonist-stimulation. We believe however
that this interesting lead requires substantial additional investigation as a follow up study,
and we now have discussed this point in the revised manuscript (page 14, line 330-334).

I can understand the need not to include every interesting follow up finding in this manuscript, but I am concerned the way the revised manuscript is written leads readers in the opposite direction from the findings the authors provided to reviewers (Figure R2.1). The relevant lines of text in the revised manuscript are as follows (lines 333-337):

vesicles¹³; however, this remains to be determined for the V2RT360A mutant in absence and presence

of Ib30. If agonist-induced endocytosis of V2RT360A also displays a pattern similar to that of β arr1
trafficking, it will be important to understand whether Ib30 rescues receptor endocytosis as well.
This may also help clarify the underlying mechanism for the lack of cAMP potentiation in case of
V2RT360A despite enhanced endosomal trafficking of β arr1 in the presence of Ib30.

The authors provided data for reviewers showing that V2RT360A DOES NOT follow a pattern of agonist induced endocytosis similar to that of bArr1. If the authors are going to speculate on this they should at least speculate about what they already know to be supported by data. If the authors are not going to include these new findings they should update this section of the text to lead readers in the direction supported by their preliminary findings.

Reviewer #3 (Remarks to the Author):

All our major points have been addressed.

Reviewer #4 (Remarks to the Author):

I would like to thank the authors for their clarification of the points that I have criticized.

My points have been adressed sufficiently.

**Response to reviewers' comments (NCOMMS-22-01576A)**

**Reviewer #1**

Almost all the recommended improvements have been done in the new version of the
manuscript, except one. It seems that the authors forgot to address the concern about the
analytic approach (see lines 62 to 67 in the rebuttal). Consequently, I recommend this
manuscript to be published conditionally to addressing this concern.

We thank the reviewer for her/his positive comments, and we apologize for missing the point
about statistical test. In fact, we have compared One Way ANOVA and t-test on these data
sets, and the overall significance including p values does not change significantly. As we are
comparing more than two groups measured concurrently in these experiments, we believe
that using One Way ANOVA for statistical analysis is appropriate.

**Reviewer #2**

The authors have done a thorough job of responding to my comments. I think the revised
manuscript is well-suited for publication. There is one author response on which I request
clarification. I can understand the need not to include every interesting follow up finding in this
manuscript, but I am concerned the way the revised manuscript is written leads readers in the
opposite direction from the findings the authors provided to reviewers. The authors provided
data for reviewers showing that V2RT360A DOES NOT follow a pattern of agonist induced
endocytosis similar to that of β arr1. If the authors are going to speculate on this, they should
at least speculate about what they already know to be supported by data. If the authors are
not going to include these new findings, they should update this section of the text to lead
readers in the direction supported by their preliminary findings.

We thank the reviewer for raising this insightful point, and following her/his suggestion, we
have modified the corresponding text accordingly (line 330-337, page 14).

**Original text:**

For example, in case of wild-type V_2R , agonist-stimulation promotes co-localization of the
receptor, β arr1 and Ib30 in endosomal vesicles¹³; however, this remains to be determined for
the V_2R^{T360A} mutant in absence and presence of Ib30. If agonist-induced endocytosis of
V_2R^{T360A} also displays a pattern similar to that of β arr1 trafficking, it will be important to
understand whether Ib30 rescues receptor endocytosis as well. This may also help clarify the
underlying mechanism for the lack of cAMP potentiation in case of V_2R^{T360A} despite enhanced
endosomal trafficking of β arr1 in the presence of Ib30.

**Revised text:**

For example, in case of wild-type V_2R , agonist-stimulation promotes co-localization of the
receptor, β arr1 and Ib30 in endosomal vesicles¹³; however, it is plausible that V_2R^{T360A}
dissociates from β arr1 at the plasma membrane due to relatively lower affinity. This may lead
to trafficking of β arr1, presumably stabilized in an active conformation by Ib30, to endosomal
vesicles even in the absence of the receptor. Further investigation along these lines in future
studies may also help clarify the underlying mechanism for the lack of cAMP potentiation in
case of V_2R^{T360A} despite enhanced endosomal trafficking of β arr1 in the presence of Ib30.

**Reviewer #3**

All our major points have been addressed.

We thank the reviewer for her/his positive comments.

**Reviewer #4**

I would like to thank the authors for their clarification of the points that I have criticized. My
points have been addressed sufficiently.

We thank the reviewer for her/his positive comments.